# ENERGY-GUIDED CONTINUOUS ENTROPIC BARYCENTER ESTIMATION FOR GENERAL COSTS

## ABSTRACT

Optimal transport (OT) barycenters are a mathematically grounded way of averaging probability distributions while capturing their geometric properties. In short, the barycenter task is to take the average of a collection of probability distributions w.r.t. given OT discrepancies. We propose a novel algorithm for approximating the continuous Entropic OT (EOT) barycenter for arbitrary OT cost functions. Our approach is built upon the dual reformulation of the EOT problem based on weak OT, which has recently gained the attention of the ML community. Beyond its novelty, our method enjoys several advantageous properties: (i) we establish quality bounds for the recovered solution; (ii) this approach seemlessly interconnects with the Energy-Based Models (EBMs) learning procedure enabling the use of well-tuned algorithms for the problem of interest; (iii) it provides an intuitive optimization scheme avoiding min-max, reinforce and other intricate technical tricks. For validation, we consider several low-dimensional scenarios and image-space setups, including *non-Euclidean* cost functions. Furthermore, we investigate the practical task of learning the barycenter on an image manifold generated by a pretrained generative model, opening up new directions for real-world applications.

## 1 INTRODUCTION

Averaging is a fundamental concept in mathematics and plays a central role in numerous applications. While it is a straightforward operation when applied to scalars or vectors in a linear space, the situation complicates when working in the space of probability distributions. Here, simple convex combinations can be inadequate or even compromise essential geometric features, which necessitates a different way of taking averages. To address this issue, one may carefully select a measure of distance that properly captures similarity in the space of probabilities. Then, the task is to find a procedure which identifies a 'center' that, on average, is closest to the reference distributions.

One good choice for comparing and averaging probability distributions is provided by the family of Optimal Transport (OT) discrepancies (Villani et al., 2009). They have clear geometrical meaning and practical interpretation (Santambrogio, 2015; Solomon, 2018). The corresponding problem of averaging probability distributions using OT discrepancies is known as the OT barycenter problem (Agueh & Carlier, 2011). OT-based barycenters find application in various practical domains: domain adaptation (Montesuma & Mboula, 2021b;a), shape interpolation (Solomon et al., 2015), Bayesian inference (Srivastava et al., 2015; 2018), text scoring (Colombo et al., 2021), style transfer (Mroueh, 2020), reinforcement learning (Metelli et al., 2019).

Over the past decade, the substantial demand from practitioners sparked the development of various methods tackling the barycenter problem. The research community's initial efforts were focused on the discrete OT barycenter setting, see Appendix B.1 for more details. The **continuous setting** turns out to be even more challenging, with only a handful of recent works devoted to this setup (Li et al., 2020; Cohen et al., 2020; Korotin et al., 2021c; 2022a; Fan et al., 2021; Noble et al., 2023; Chi et al., 2023). Most of these works are devoted to specific OT cost functions, e.g., deal with $\ell_2^2$ barycenters (Korotin et al., 2021c; 2022a; Fan et al., 2021; Noble et al., 2023); while others require non-trivial *a priori* selections (Li et al., 2020) and have limiting expressivity and generative ability (Cohen et al., 2020; Chi et al., 2023), see §3 for a detailed discussion.

**Contribution**. We propose a novel approach for solving Entropy-regularized OT (EOT) barycenter problems, which alleviates the aforementioned limitations of existing continuous OT solvers.

- We reveal an elegant reformulation of the EOT barycenter problem by combining weak dual form of EOT with the congruence condition (§4.1); we derive a simple optimization procedure which closely relates to the standard training algorithm of Energy-Based models (EBMs) (§4.2).

- We establish the generalization bounds as well as the universal approximation guarantees for our recovered EOT plans which push the reference distributions to the barycenter (§4.3).

- We validate the applicability of our approach on various toy and large-scale setups including the RGB image domain (§5). In contrast to previous works, we also pay attention to non-Euclidean OT costs. Specifically, we conduct a series of experiments looking for a barycenter on an image manifold of a pretrained GAN. In principle, the image manifold support may contribute to the interpretability and plausibility of the resulting barycenter distribution in downstream tasks.

**Notations.** We write $\overline{K} = \{1, 2, \ldots, K\}$. Throughout the paper $\mathcal{X} \subset \mathbb{R}^{D'}$, $\mathcal{Y} \subset \mathbb{R}^D$ and $\mathcal{X}_k \subset \mathbb{R}^{D_k}$ ($k \in \overline{K}$) are compact subsets of Euclidean spaces. Continuous functions on $\mathcal{X}$ are denoted as $\mathcal{C}(\mathcal{X})$. Probability distributions on $\mathcal{X}$ are $\mathcal{P}(\mathcal{X})$. Absolutely continuous probability distributions on $\mathcal{X}$ are denoted by $\mathcal{P}_{ac}(\mathcal{X}) \subset \mathcal{P}(\mathcal{X})$. Given $\mathbb{P} \in \mathcal{P}(\mathcal{X}), \mathbb{Q} \in \mathcal{P}(\mathcal{Y})$, we use $\Pi(\mathbb{P}, \mathbb{Q})$ to designate the set of *transport plans*, i.e., probability distributions on $\mathcal{X} \times \mathcal{Y}$ with the first and second marginals given by $\mathbb{P}$ and $\mathbb{Q}$, respectively. The density of $\mathbb{P} \in \mathcal{P}_{ac}(\mathcal{X})$ w.r.t. the Lebesgue measure is denoted by $\frac{d\mathbb{P}(x)}{dx}$.

## 2 BACKGROUND

First, we recall the formulations of EOT (§2.1) and the barycenter problem (§2.2). Subsequently, we clarify the computational setup of the considered EOT barycenter problem (§2.3).

### 2.1 ENTROPIC OPTIMAL TRANSPORT

Consider distributions $\mathbb{P} \in \mathcal{P}_{ac}(\mathcal{X})$, $\mathbb{Q} \in \mathcal{P}_{ac}(\mathcal{Y})$, a continuous cost function $c : \mathcal{X} \times \mathcal{Y} \to \mathbb{R}$ and a regularization parameter $\varepsilon > 0$. The *entropic optimal transportation* (EOT) problem between $\mathbb{P}$ and $\mathbb{Q}$ (Cuturi, 2013; Peyré et al., 2019; Genevay, 2019) consists of finding a minimizer of

$$\text{EOT}_{c,\varepsilon}^{(2)}(\mathbb{P}, \mathbb{Q}) \overset{\text{def}}{=} \min_{\pi \in \Pi(\mathbb{P}, \mathbb{Q})} \int_{\mathcal{X} \times \mathcal{Y}} c(x, y) d\pi(x, y) - \varepsilon H(\pi), \tag{1}$$

where $H(\pi)$ is the differential entropy of plan $\pi$. The case $\varepsilon = 0$ corresponds to classical OT, also known as the Kantorovich problem (Kantorovich, 1942), and falls of the scope of this paper. Since by the chain rule of entropy we have $H(\pi) = \int_{\mathcal{X}} H(\pi(\cdot|x)) d\mathbb{P}(x) + H(\mathbb{P})$, where $\pi(\cdot|x)$ are conditional distributions on $\mathcal{Y}$, equation (1) permits the following equivalent reformulation:

$$\text{EOT}_{c,\varepsilon}(\mathbb{P}, \mathbb{Q}) \overset{\text{def}}{=} \min_{\pi \in \Pi(\mathbb{P}, \mathbb{Q})} \int_{\mathcal{X} \times \mathcal{Y}} c(x, y) d\pi(x, y) - \varepsilon \int_{\mathcal{X}} H(\pi(\cdot|x)) \underbrace{d\mathbb{P}(x)}_{=d\pi(x)}, \tag{2}$$

A minimizer $\pi^* \in \Pi(\mathbb{P}, \mathbb{Q})$ of (2) is called the EOT plan. Thanks to the equivalence of (1) and (2), its existence and uniqueness are guaranteed, see, e.g., (Clason et al., 2021, Th. 3.3). In practice, we usually do not require the EOT plan $\pi^*$ but its conditional distributions $\pi^*(\cdot|x) \in \mathcal{P}(\mathcal{Y})$ as they prescribe how points $x \in \mathcal{X}$ are stochastically mapped to $\mathcal{Y}$ (Gushchin et al., 2023b, §2). We refer to $\pi^*(\cdot|x)$ as the *conditional plans* (for $x \in \mathcal{X}$).

**Weak OT dual formulation of the EOT problem**. The EOT problem permits several dual formulations. In our paper, we use the one derived from the weak OT theory, see (Gozlan et al., 2017, Theorem 9.5) or (Backhoff-Veraguas et al., 2019, Theorem 1.3):

$$\text{EOT}_{c,\varepsilon}(\mathbb{P}, \mathbb{Q}) = \sup_{f \in \mathcal{C}(\mathcal{Y})} \left\{ \int_{\mathcal{X}} f^C(x) d\mathbb{P}(x) + \int_{\mathcal{Y}} f(y) d\mathbb{Q}(y) \right\}, \tag{3}$$

where $f^C : \mathcal{X} \to \mathbb{R}$ is the so-called **weak** entropic $c$-transform (Backhoff-Veraguas et al., 2019, Eq. 1.2) of the function (*potential*) $f$. The transform is defined by

$$f^C(x) \overset{\text{def}}{=} \min_{\mu \in \mathcal{P}(\mathcal{Y})} \left\{ \int_{\mathcal{Y}} c(x, y) d\mu(y) - \varepsilon H(\mu) - \int_{\mathcal{Y}} f(y) d\mu(y) \right\}. \tag{4}$$

We use the capital $C$ in $f^C$ to distinguish the weak transform from the classic $c$-transform (Santambrogio, 2015, §1.6) or $(c, \epsilon)$-transform (Marino & Gerolin, 2020, §2). In particular, formulation (3) is not to be confused with the conventional dual, see (Mokrov et al., 2023, Appendix A).

For each $x \in \mathcal{X}$, the minimizer $\mu_x^f \in \mathcal{P}(\mathcal{Y})$ of the weak $c$-transform (4) exists and is unique. Its density is of the form (Mokrov et al., 2023, Theorem 1): Set $Z_c(f, x) \stackrel{\text{def}}{=} \int_{\mathcal{Y}} \exp\left(\frac{f(y)-c(x,y)}{\varepsilon}\right) \mathrm{d}y$,

$$\frac{\mathrm{d}\mu_x^f(y)}{\mathrm{d}y} \stackrel{\text{def}}{=} \frac{1}{Z_c(f, x)} \exp\left(\frac{f(y) - c(x, y)}{\varepsilon}\right). \tag{5}$$

By substituting (5) into (4) and carrying out straightforward mathematical manipulations, we arrive at an explicit formula $f^C(x) = -\varepsilon \log Z_c(f, x)$, see (Mokrov et al., 2023, Equation (14)).

Not only does maximizing the dual objective in (3) allow us to estimate the actual value of the primal objective $\text{EOT}_{c,\varepsilon}(\mathbb{P}, \mathbb{Q})$, but it also provides an approximation of the EOT plan $\pi^*$. Consider the distribution $\mathrm{d}\pi^f(x, y) = \mathrm{d}\mu_x^f(y)\mathrm{d}\mathbb{P}(x)$. According to (Mokrov et al., 2023, Thm. 2), we have

$$\text{EOT}_{c,\varepsilon}(\mathbb{P}, \mathbb{Q}) - \left( \int_{\mathcal{X}} f^C(x)\mathrm{d}\mathbb{P}(x) + \int_{\mathcal{Y}} f(y)\mathrm{d}\mathbb{Q}(y) \right) = \varepsilon \text{KL}\left(\pi^* \| \pi^f\right). \tag{6}$$

This means that the smaller the error in solving the dual problem with $f \in \mathcal{C}(\mathcal{Y})$, the closer the distribution $\pi^f$ to $\pi^*$. This useful property appears only when $\epsilon > 0$, which is one of many reasons why EOT ($\epsilon > 0$) is often favoured over the unregularized problem ($\epsilon = 0$).

## 2.2 ENTROPIC OT BARYCENTER

Consider distributions $\mathbb{P}_k \in \mathcal{P}_{\text{ac}}(\mathcal{X}_k)$ and continuous cost functions $c_k(\cdot, \cdot) : \mathcal{X}_k \times \mathcal{Y} \to \mathbb{R}$ for $k \in \overline{K}$. Given positive weights $\lambda_k > 0$ with $\sum_{k=1}^K \lambda_k = 1$, the EOT Barycenter problem (Cuturi & Doucet, 2014; Cuturi & Peyré, 2018; Dvurechenskii et al., 2018; del Barrio & Loubes, 2020), (Le et al., 2021; 2022) is:

$$\inf_{\mathbb{Q} \in \mathcal{P}(\mathcal{Y})} \sum_{k=1}^K \lambda_k \text{EOT}_{c_k,\varepsilon}^{(2)}(\mathbb{P}_k, \mathbb{Q}). \tag{7}$$

The case where $\varepsilon = 0$ corresponds to the classical OT barycenter problem (Agueh & Carlier, 2011) and falls of the scope of this paper. By substituting $\text{EOT}_{c_k,\varepsilon}^{(2)}$, with $\text{EOT}_{c_k,\varepsilon}$ we derive

$$\mathcal{L}^* \stackrel{\text{def}}{=} \inf_{\mathbb{Q} \in \mathcal{P}(\mathcal{Y})} \mathcal{B}(\mathbb{Q}) \stackrel{\text{def}}{=} \inf_{\mathbb{Q} \in \mathcal{P}(\mathcal{Y})} \sum_{k=1}^K \lambda_k \text{EOT}_{c_k,\varepsilon}(\mathbb{P}_k, \mathbb{Q}) \tag{8}$$

which differs from (7) only by the additive constant $\sum_{k=1}^K \lambda_k H(\mathbb{P}_k)$ and has the **same minimizers**. It is worth noting that the functional $\mathbb{Q} \mapsto \mathcal{B}(\mathbb{Q})$ is strictly convex and lower semicontinuous (w.r.t. the weak topology) as each component $\mathbb{Q} \mapsto \text{EOT}_{c_k,\varepsilon}(\mathbb{P}_k, \mathbb{Q})$ is strictly convex and l.s.c. (lower semi-continuous) itself. The latter follows from (Backhoff-Veraguas et al., 2019, Th. 2.9) by noting that on $\mathcal{P}(\mathcal{Y})$ the map $\mu \mapsto \int_{\mathcal{Y}} c_k(x, y)\mathrm{d}\mu(y) - H(\mu)$ is l.s.c, bounded from below and strictly convex thanks to the entropy term. Since $\mathcal{P}(\mathcal{Y})$ is weakly compact (as $\mathcal{Y}$ is compact due to Prokhorov's theorem, see, e.g., (Santambrogio, 2015, Box 1.4)), it holds that $\mathcal{B}(\mathbb{Q})$ admits at least one minimizer due to the Weierstrass theorem (Santambrogio, 2015, Box 1.1), i.e., a barycenter $\mathbb{Q}^*$ exists. In the paper, *we work under the reasonable assumption that there exists at least one $\mathbb{Q}$ for which $\mathcal{B}(\mathbb{Q}) < \infty$*. In this case, the barycenter $\mathbb{Q}^*$ is unique as consequence of the strict convexity of $\mathcal{B}$.

## 2.3 COMPUTATIONAL ASPECTS OF THE EOT BARYCENTER PROBLEM

Barycenter problems, such as (7) or (8), are known to be challenging in practice (Altschuler & Boix-Adsera, 2022). To our knowledge, even when $\mathbb{P}_1, \ldots, \mathbb{P}_K$ are Gaussian distributions, there is no direct analytical solution neither for our entropic case ($\epsilon > 0$, see the additional discussion in Appendix C.4, nor for the unregularized case (Álvarez-Esteban et al., 2016, $\epsilon = 0$). Furthermore, in real-world scenarios, the distributions $\mathbb{P}_k$ ($k \in \overline{K}$) are typically not available explicitly but only through empirical samples (datasets). This aspect leads to the following **learning setup**.

> We assume that each $\mathbb{P}_k$ is accessible only by a limited number of i.i.d. empirical samples $X_k = \{x_k^1, x_k^2, \ldots x_k^{N_k}\} \sim \mathbb{P}_k$. Our aim is to approximate the optimal conditional plans $\pi_k^*(\cdot|x_k)$ between the entire source distributions $\mathbb{P}_k$ and the entire (unknown) barycenter $\mathbb{Q}^*$ solving (8). The recovered plans should provide the *out-of-sample* estimation, i.e., allow generating samples from $\pi_k^*(\cdot|x_k^{\text{new}})$, where $x_k^{\text{new}}$ is a new sample from $\mathbb{P}_k$ which is not necessarily present in the train sample.

This setup corresponds to **continuous OT** (Li et al., 2020; Korotin et al., 2021c). It differs from the **discrete OT** setup (Cuturi, 2013; Cuturi & Doucet, 2014) which aims to solve the barycenter problem for empirical distributions, e.g., $\widehat{\mathbb{P}}_k \stackrel{\text{def}}{=} \frac{1}{N_k} \sum_{n=1}^{N_k} \delta_{x_k^n}$. Discrete methods are not well-suited for the out-of-sample estimation required in the continuous OT setup.

## 3 RELATED WORKS

The taxonomy of OT solvers is large. Due to space constraints, we discuss here only methods within the *continuous OT learning setup* that solve the EOT problem or (E-)OT barycenter problem. These methods try to approximate OT maps or plans between the distributions $\mathbb{P}_k$ and the barycenter $\mathbb{Q}^*$ rather than just their empirical counterparts that are available from the training samples. A broader discussion of general-purpose discrete and continuous OT solvers is left to Appendix B.1.

**Continuous EOT solvers** aim to recover the optimal EOT plan $\pi^*$ in (2), (1) between unknown distributions $\mathbb{P}$ and $\mathbb{Q}$ which are only accessible through a limited number of samples. One group of methods (Genevay et al., 2016; Seguy et al., 2018; Daniels et al., 2021) is based on the dual formulation of OT problem regularized with KL divergences (Genevay et al., 2016, Eq. ($\mathcal{P}_\varepsilon$)) which is equivalent to (2). Another group of methods (Vargas et al., 2021; De Bortoli et al., 2021; Chen et al., 2021; Gushchin et al., 2023a; Tong et al., 2023; Shi et al., 2023) takes advantage of the dynamic reformulation of (1) via Schrödinger bridges (Léonard, 2013; Marino & Gerolin, 2020).

In (Mokrov et al., 2023), the authors propose an approach to tackle (2) by means of Energy-Based models (LeCun et al., 2006; Song & Kingma, 2021, EBM). They develop an optimization procedure resembling standard EBM training which retrieves the optimal dual potential $f^*$ appearing in (3). As a byproduct, they recover the optimal conditional plans $\mu_x^{f^*} = \pi^*(\cdot|x)$. Our approach for solving the EOT barycenter (8) is primarily inspired by this work. In fact, we manage to overcome the theoretical and practical difficulties that arise when moving from the EOT problem guided with EBMs to the EOT barycenter problem (multiple marginals, optimization with respect to an *unfixed* marginal distribution $\mathbb{Q}$), see §4 for details of our method.

**Continuous OT barycenter solvers** are based on the un-regularized or regularized OT barycenter problem within the continuous OT learning setup. The works (Korotin et al., 2021c; Fan et al., 2021; Noble et al., 2023; Korotin et al., 2022a) are designed *exclusively* for the quadratic Euclidean cost $\ell^2(x, y) \stackrel{\text{def}}{=} \frac{1}{2}\|x - y\|_2^2$. The OT

| Method | Admissible OT costs | Learns OT plans | Max considered data dim | Regularization |
|--------|--------------------|-----------------|------------------------|----------------|
| (Li et al., 2020) | general | yes | 8D, no images | Entropic/Quadratic with *fixed* prior |
| (Cohen et al., 2020) | general | no | 1x32x32 (MNIST) | Entropic (Sinkhorn) |
| (Korotin et al., 2021c) | only $l_2^2$ | yes | 256D, no images | requires *fixed* prior |
| (Fan et al., 2021) | only $l_2^2$ | yes | 1x28x28 (MNIST) | no |
| (Korotin et al., 2022a) | only $l_2^2$ | yes | 3x64x64 (CelebA, etc.) | no |
| (Noble et al., 2023) | only $l_2^2$ | yes | 1x28x28 (MNIST) | Entropic |
| (Chi et al., 2023) | general | yes | 256D, Gaussians only | Entropic/Quadratic |
| **Ours** | general | yes | 3x64x64 (CelebA) | Entropic |

Table 1: Comparison of continuous OT barycenter solvers

problem with this particular cost exhibits several advantageous theoretical properties (Ambrosio & Gigli, 2013, §2) which are exploited by the aforementioned articles to build efficient procedures for barycenter estimation algorithms. In particular, (Korotin et al., 2021c; Fan et al., 2021) utilize ICNNs (Amos et al., 2017) which parameterize convex functions, and (Noble et al., 2023) relies on a specific tree-based Schrödinger Bridge reformulation. In contrast, our proposed approach is designed to handle the EOT problem with *arbitrary* cost functions $c_1, \ldots, c_K$. In (Li et al., 2020), they also consider regularized OT with non-Euclidean costs. Similar to our method, they take advantage of the dual formulation and exploit the so-called congruence condition (§4). However, their optimization procedure substantially differs. It necessitates selecting a *fixed prior* for the barycenter, which can be non-trivial. The work (Chi et al., 2023) takes a step further by directly optimizing the barycenter distribution in a variational manner, eliminating the need for a *fixed prior*. This modification increases the complexity of optimization and requires specific parametrization of the variational barycenter. In (Cohen et al., 2020), the authors also parameterize the barycenter as a generative model. Their approach does not recover the OT plans, which differs from our learning setup (§2.3). A summary of the key properties is provided in Table 1, highlighting the fact that our proposed approach overcomes many imperfections which are inherent in competing methods.

## 4 PROPOSED BARYCENTER SOLVER

In the first two subsections, we work out our optimization objective (§4.1) and its practical implementation (§4.2). In §4.3, we alleviate the gap between the theory and practice by offering finite sample approximation guarantees and universality of NNs to approximate the solution.

### 4.1 DERIVING THE OPTIMIZATION OBJECTIVE

In what follows, we analyze (8) from the dual perspectives. We introduce $\mathcal{L} : \mathcal{C}(\mathcal{Y})^K \to \mathbb{R}$:

$$\mathcal{L}(f_1, \ldots, f_K) \stackrel{\text{def}}{=} \sum_{k=1}^{K} \lambda_k \int_{\mathcal{X}_k} f_k^{C_k}(x_k) \mathrm{d}\mathbb{P}_k(x_k) \qquad \left[ = -\epsilon \sum_{k=1}^{K} \lambda_k \int_{\mathcal{X}_k} \log Z_{c_k}(f_k, x_k) \mathrm{d}\mathbb{P}_k(x_k) \right].$$

Here $f_k^{C_k}(x_k)$ denotes the weak entropic $c_k$-transform (4) of $f_k$. Following §2.1, we see that it coincides with $-\epsilon \log Z_{c_k}(f_k, x_k)$. Below we formulate our main theoretical result, which will allow us to solve the EOT barycenter task without optimization over all probablity measures on $\mathcal{Y}$.

**Theorem 1** (Dual formulation of the EOT barycenter problem). *The EOT barycenter problem* (8) *permits the following dual formulation:*

$$\inf_{\mathbb{Q} \in \mathcal{P}(\mathcal{Y})} \sum \lambda_k \mathrm{EOT}_{c_k, \varepsilon}(\mathbb{P}_k, \mathbb{Q}) = \mathcal{L}^* = \sup_{\substack{f_1, \ldots, f_K \in \mathcal{C}(\mathcal{Y}); \\ \sum_{k=1}^{K} \lambda_k f_k = 0}} \mathcal{L}(f_1, \ldots, f_K). \qquad (9)$$

We refer to the constraint $\sum_{k=1}^{K} \lambda_k f_k = 0$ as the **congruence** condition w.r.t. weights $\{\lambda_k\}_{k=1}^{K}$. The potentials $f_k$ appearing in (9) play the same role as in (3). Notably, when $\mathcal{L}(f_1, \ldots, f_K)$ is close to $\mathcal{L}^*$, the conditional optimal transport plans $\pi_k^*(\cdot|x_k), x_k \in \mathcal{X}_k$, between $\mathbb{P}_k$ and the barycenter distribution $\mathbb{Q}^*$ can be approximately recovered through the potentials $f_k$. This intuition is formalized in Theorem 2 below. First, for $f_k \in \mathcal{C}(\mathcal{Y})$, we define

$$\mathrm{d}\pi^{f_k}(x_k, y) \stackrel{\text{def}}{=} \mathrm{d}\mu_{x_k}^{f_k}(y) \mathrm{d}\mathbb{P}_k(x_k) \qquad \text{and} \qquad \mathrm{d}\mathbb{Q}^{f_k}(y) \stackrel{\text{def}}{=} \int_{\mathcal{X}_k} \mathrm{d}\pi^{f_k}(x_k, y).$$

**Theorem 2** (Quality bound of plans recovered from dual potentials). *Let* $\{f_k\}_{k=1}^{K}, f_k \in \mathcal{C}(\mathcal{Y})$ *be congruent potentials. Then we have*

$$\mathcal{L}^* - \mathcal{L}(f_1, \ldots, f_K) = \epsilon \sum_{k=1}^{K} \lambda_k KL\left(\pi_k^* \| \pi^{f_k}\right) \geq \epsilon \sum_{k=1}^{K} \lambda_k KL\left(\mathbb{Q}^* \| \mathbb{Q}^{f_k}\right), \qquad (10)$$

*where* $\pi_k^* \in \Pi(\mathbb{P}_k, \mathbb{Q}^*), k \in \overline{K}$ *are the EOT plans between* $\mathbb{P}_k$ *and the barycenter distribution* $\mathbb{Q}^*$.

According to Theorem 2, an approximate solution $\{f_k\}_{k=1}^{K}$ of (9) yields distributions $\pi^{f_k}$ which are close to the optimal plans $\pi_k^*$. Each $\pi^{f_k}$ is formed by conditional distributions $\mu_{x_k}^{f_k}$, c.f. (5), with closed-form energy function, i.e., the unnormalized log-likelihood. Consequently, one can generate samples from $\mu_{x_k}^{f_k}$ using standard MCMC techniques (Andrieu et al., 2003). In the next subsection, we stick to the practical aspects of optimization of (9), which bears certain similarities to the training of Energy-Based models (LeCun et al., 2006; Song & Kingma, 2021, EBM).

**Relation to prior works.** Works (Li et al., 2020; Korotin et al., 2021c) also aim to first get the dual potentials and then recover the barycenter, see the detailed discussion in §3.

## 4.2 PRACTICAL OPTIMIZATION ALGORITHM

To maximize the dual EOT barycenter objective (9), we replace the potentials $f_k \in \mathcal{C}(\mathcal{Y})$ for $k \in \overline{K}$ with neural networks $f_{\theta,k}$, $\theta \in \Theta$. In order to eliminate the constraint in (9), we propose parametrizing $f_{\theta,k}$ as $g_{\theta_k} - \sum_{k'=1}^{K} \lambda_{k'} g_{\theta_{k'}}$, where $\{g_{\theta_k} : \mathbb{R}^D \to \mathbb{R}, \theta_k \in \Theta_k\}_{k=1}^{K}$ are neural networks. This parameterization automatically ensures the congruence condition $\sum_{k=1}^{K} \lambda_k f_{\theta,k} \equiv 0$. Note that $\Theta = \Theta_1 \times \cdots \times \Theta_K$ and $\theta = (\theta_1, \ldots, \theta_K) \in \Theta$. Our objective function for (9) is defined as

$$L(\theta) \stackrel{\text{def}}{=} -\varepsilon \sum_{k=1}^{K} \lambda_k \int_{\mathcal{X}_k} \log Z_{c_k}(f_{\theta,k}, x_k) \mathrm{d}\mathbb{P}_k(x_k). \qquad (11)$$

While direct computation of the normalizing constant $Z_{c_k}$ may, in general, be infeasible, the gradient of $L$ with respect to $\theta$ can be derived similarly to (Mokrov et al., 2023, Theorem 3):

**Theorem 3** (Gradient of the dual EOT barycenter objective). *The gradient of $L$ satisfies*

$$\frac{\partial}{\partial \theta} L(\theta) = -\sum_{k=1}^{K} \lambda_k \left\{ \int_{\mathcal{X}_k} \int_{\mathcal{Y}} \left[ \frac{\partial}{\partial \theta} f_{\theta,k}(y) \right] \mathrm{d}\mu_{x_k}^{f_{\theta,k}}(y) \mathrm{d}\mathbb{P}_k(x_k) \right\}. \qquad (12)$$

With this result, we can describe our proposed algorithm which maximizes $L$ using (12).

TRAINING. To perform stochastic gradient ascent step w.r.t. $\theta$, we approximate (12) with Monte-Carlo by drawing samples from $\mathrm{d}\pi^{f_{\theta,k}}(x_k, y) = \mathrm{d}\mu_{x_k}^{f_{\theta,k}}(y) \mathrm{d}\mathbb{P}_k(x_k)$. Analogously to (Mokrov et al., 2023, §3.2), this can be achieved by a simple two-stage procedure. At first, we draw a random vector $x_k$ from $\mathbb{P}_k$. This is done by picking a random empirical sample from the available dataset

---

**Algorithm 1:** Entropic OT barycenters via Energy-Based Modelling

---

**Input** : Source distributions $\mathbb{P}_k$, $k \in \overline{K}$ accessible by samples;
cost functions $c_k(x_k, y) : \mathcal{X}_k \times \mathcal{Y} \to \mathbb{R}$; the regularization coefficient $\varepsilon > 0$;
barycenter averaging coefficients $\lambda_k > 0$ such that $\sum_{k=1}^{K} \lambda_k = 1$;
MCMC procedure `MCMC_procedure`; batch size $S > 0$;
potential NNs $f_{\theta,k} : \mathcal{Y} \to \mathbb{R}$, such that $\sum_{k=1}^{K} \lambda_k f_{\theta,k} \equiv 0$ (see §4.2).

**Output:** trained NNs $f_{\theta^*,k}$ recovering the conditional OT plans between $\mathbb{P}_k$ and barycenter $\mathbb{Q}^*$.

**for** $iter = 1, 2, \ldots$ **do**

   **for** $k = 1, 2, \ldots, K$ **do**

      Sample batch $\{x_k^s\}_{s=1}^{S} \sim \mathbb{P}_k$;

      Draw $Y_k = \{y_k^s\}_{s=1}^{S}$ with MCMC: $y_k^s = $ `MCMC_procedure` $\left( \frac{f_{\theta,k}(\cdot) - c(x_k^s, \cdot)}{\varepsilon} \right)$;

      $\widehat{L}_k \leftarrow -\lambda_k \frac{1}{S} \left[ \sum_{s=1}^{S} f_{\theta,k}(y_k^s) \right]$;

   $\widehat{L} \leftarrow \sum_{k=1}^{K} \widehat{L}_k$; Perform a gradient step over $\theta$ by using $\frac{\partial \widehat{L}}{\partial \theta}$;

---

$X_k$. Then, we need to draw a sample from the distribution $\mu_{x_k}^{f_{\theta,k}}$. Since we know the negative **energy** (unnormalized log density) of $\mu_{x_k}^{f_{\theta,k}}$ by (5), we can sample from this distribution by applying an MCMC procedure which uses the negative energy function $\varepsilon^{-1}(f_{\theta,k}(y) - c_k(x_k, y))$ as the input. Our findings are summarized in Algorithm 1.

In all our experiments, we use ULA (Roberts & Tweedie, 1996, §1.4.1) as a `MCMC_procedure`. It is the simplest MCMC algorithm. Specifically, in order to draw a sample $y_k \sim \mu_{x_k}^{f_{\theta,k}}$, where $x_k \in \mathcal{X}_k$, we initialize $y_k^{(0)}$ from the standard Normal $D$−dimensional distribution $\mathcal{N}(0, I_D)$ and then iterate the discretized Langevin dynamics:

$$y_k^{(l+1)} \leftarrow y_k^{(l)} + \frac{\eta}{2\varepsilon} \nabla_y \big( f_{\theta,k}(y) - c(x_k, y) \big) \Big|_{y=y_k^{(l)}} + \sqrt{\eta}\xi_l, \quad \xi_l \sim \mathcal{N}(0, I_D),$$

where $l \in \{0, 1, 2, \ldots, L\}$, $L$ is a number of steps, and $\eta > 0$ is a step size. Note that the iteration procedure above could be straightforwardly adapted to a batch scenario, i.e., we can simultaneously simulate the whole batch of samples $Y_k^{(l)}$ conditioned on $X_k^{(l)}$. The particular values of number of steps $L$ and step size $\eta$ are reported in the details of the experiments, see Appendix C. An alternative importance sampling-based approach for optimizing (12) is presented in Appendix D.

INFERENCE. At the inference stage, we use the same ULA procedure for sampling from the recovered optimal conditional plans $\pi^{f_{\theta^*,k}}(\cdot|x_k)$, see the details on the hyperparameters $L, \eta$ in §5.

**Relation to prior works.** Learning a distribution of interest via its energy function (EBMs) is a well-established direction in generative modelling research (LeCun et al., 2006; Xie et al., 2016; Du & Mordatch, 2019; Song & Kingma, 2021). Similar to ours, the key step in most energy-based approaches is the MCMC procedure which recovers samples from a distribution accessible only by an unnormalized log density. Typically, various techniques are employed to improve the stability and convergence speed of MCMC, see, e.g., (Du et al., 2021; Gao et al., 2021; Zhao et al., 2021). The majority of these techniques can be readily adapted to complement our approach. At the same time, the primary goal of this study is to introduce and validate the methodology for computing EOT barycenters in an energy-guided manner. Therefore, we opt for the **simplest** MCMC algorithm, even **without the replay buffer** (Hinton, 2002), as it serves our current objectives.

### 4.3 GENERALIZATION BOUNDS AND UNIVERSAL APPROXIMATION WITH NEURAL NETS

In this subsection, we answer the question of how far the recovered plans are from the EOT plan $\pi_k^*$ between $\mathbb{P}_k$ and $\mathbb{Q}$. In practice, for each distribution $\mathbb{P}_k$ we know only the empirical samples $X_k = \{x_k^1, x_k^2, \ldots x_k^{N_k}\} \sim \mathbb{P}_k$, i.e., finite datasets. Besides, the available potentials $f_k$, $k \in \overline{K}$ come from restricted classes of functions and satisfy the congruence condition. More precisely, we have $f_k = g_k - \sum_{k=1}^{K} \lambda_k g_k$ (§4.2), where each $g_k$ is picked from some class $\mathcal{G}_k$ of neural networks. Formally, we write $(f_1, \ldots, f_K) \in \overline{\mathcal{F}}$ to denote the congruent potentials constructed this way from the functional classes $\mathcal{G}_1, \ldots, \mathcal{G}_K$. Hence, in practice, we optimize the *empirical version* of (11):

$$\max_{(f_1,\ldots,f_K)\in\overline{\mathcal{F}}} \widehat{\mathcal{L}}(f_1,\ldots,f_K) \stackrel{\text{def}}{=} \max_{(f_1,\ldots,f_K)\in\overline{\mathcal{F}}} \sum_{k=1}^{K} \frac{\lambda_k}{N_k} \sum_{n=1}^{N_k} f_k^{C_k}(x_k^n) \qquad (13)$$

and recover $(\widehat{f_1},\ldots\widehat{f_K}) \stackrel{\text{def}}{=} \arg\max_{(f_1,\ldots,f_K)\in\overline{\mathcal{F}}} \widehat{\mathcal{L}}(f_1,\ldots,f_k)$. A natural question arises: ***How close are the recovered plans*** $\pi^{\widehat{f_k}}$ ***to the EOT plans*** $\pi_k^*$ between $\mathbb{P}_k$ and $\mathbb{Q}^*$? Since the objective (11) is a sum of integrals over distributions $\mathbb{P}_k$, we can derive generic finite learning guarantees by using Theorem 2, see the next theorem below.

**Theorem 4** (Finite sample learning guarantees). *The following generalization bound holds true:*

$$\epsilon\mathbb{E}\sum_{k=1}^{K}\lambda_k KL\left(\pi_k^*\|\pi^{\widehat{f_k}}\right) \leq \overbrace{4\sum_{k=1}^{K}\lambda_k\mathcal{R}_{N_k}(\mathcal{F}_k^{C_k},\mathbb{P}_k)}^{\text{Estimation error (upper bound)}} + \overbrace{\left[\mathcal{L}^* - \max_{(f_1,\ldots,f_K)\in\overline{\mathcal{F}}}\mathcal{L}(f_1,\ldots,f_K)\right]}^{\text{Approximation error}}, \qquad (14)$$

*where* $\mathcal{F}_k^{C_k} \stackrel{\text{def}}{=} \{f_k^{C_k} \mid (f_1,\ldots,f_K)\in\overline{\mathcal{F}}\}$, *and the expectation is taken w.r.t. the random realization of the datasets* $X_1{\sim}\mathbb{P}_1,\ldots,X_K{\sim}\mathbb{P}_K$. *Here* $\mathcal{R}_{N_k}(\mathcal{F}^{C_k},\mathbb{P}_k)$ *is the Rademacher complexity (Shalev-Shwartz & Ben-David, 2014, §26) of the functional class* $\mathcal{F}_k^{C_k}$ *w.r.t. samples of size* $N_k$ *from* $\mathbb{P}_k$.

While the estimation error usually decreases when the sample sizes tend to infinity, it is natural to wonder whether the approximation error can be also made arbitrarily small. We positively answer this question when a neural network parameterization is used for the potentials (as we do in §4.2).

**Theorem 5** (Universal approximation with neural networks). *For every* $\delta > 0$ *there exist* $K$ *neural networks* $g_k : \mathbb{R}^{D_k} \to \mathbb{R}$ *such that the congruent functions* $f_k = g_k - \sum_{k=1}^{K}\lambda_k g_k$ *satisfy* $\epsilon\sum_{k=1}^{K}\lambda_k KL\left(\pi_k^*\|\pi^{f_k}\right) = \mathcal{L}^* - \mathcal{L}(f_1,\ldots,f_K) < \delta$.

**Relation to prior works.** To our knowledge, the generalization and the universal approximation are novel results with no analogs established for any other continuous barycenter solver. Our analysis shows that the EOT barycenter objective (11) is well-suited for statistical learning and approximation theory tools. This aspect distinguishes our work from the preceding works, where more complex optimization objectives may not be as amenable to rigorous study.

## 5 EXPERIMENTS

We assess the performance of our barycenter solver on small-dimensional illustrative setups (§5.1) and in image spaces (§5.2, §5.3). The source code for our solver is available in the supplementary material and written in the PyTorch framework. The code will be made publicly available. The experiments are issued in the form of convenient *.ipynb notebooks. Reproducing the most challenging experiments (§5.2, §5.3) requires less than 12 hours on a single TeslaV100 GPU. The details of the experiments and extended experimental results are in Appendix C.

**Disclaimer.** Evaluating how well our solver recovers the EOT barycenter is challenging because the ground truth barycenter is typically unknown. In some cases, the true *unregularized* barycenter ($\epsilon = 0$) can be derived (see below). The EOT barycenter for sufficiently small $\epsilon > 0$ is expected to be close to the unregularized barycenter. Therefore, in most cases, our evaluation strategy is to compare the computed EOT barycenter (for small $\epsilon$) with the unregularized one. In particular, we follow this strategy to quantitatively evaluate our solver in the Gaussian case, see Appendix C.4.

### 5.1 BARYCENTERS OF 2D DISTRIBUTIONS

**2D Twister**. Consider a map $u : \mathbb{R}^2 \to \mathbb{R}^2$ which, in the *polar coordinate system*, is represented by $\mathbb{R}_+ \times [0, 2\pi) \ni (r, \theta) \mapsto \left(r, (\theta - r) \bmod 2\pi\right)$. Let $\mathbb{P}_1, \mathbb{P}_2, \mathbb{P}_3$ be 2-dimensional distributions as shown in Fig. 1a. For these distributions and uniform weights $\lambda_k = \frac{1}{3}$, the unregularized barycenter ($\epsilon = 0$) for the **twisted** cost $c_k(x_k, y) = \frac{1}{2}\|u(x_k) - u(y)\|^2$ can be derived analytically, see Appendix C.1. The barycenter is the centered Gaussian distribution which is also shown in Fig. 1a. We run the experiment for this cost with $\epsilon = 10^{-2}$, and the results are recorded in Fig. 1b. We see that it qualitatively coincides with the true barycenter.

For completeness, we also show the EOT barycenter computed with our solver for $\ell^2(x, y) = \frac{1}{2}\|x - y\|^2$ costs (Fig. 1c) and the same regularization parameter $\epsilon$. The true $\ell^2$ barycenter is estimated by using the free_support_barycenter solver from POT package (Flamary et al., 2021). We stress that the twisted cost barycenter and $\ell^2$ barycenter differ, and so do the learned conditional plans. To be precise, the $\ell^2$ EOT plan (Fig. 1d) expectedly looks more well-structured while for the twisted cost (Fig. 1b) it becomes more chaotic due to non-trivial structure of this cost.

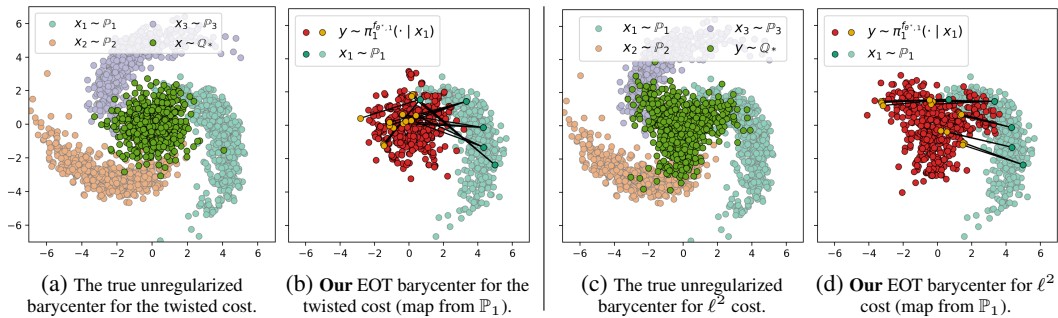

(a) The true unregularized barycenter for the twisted cost.

(b) **Our** EOT barycenter for the twisted cost (map from $\mathbb{P}_1$).

(c) The true unregularized barycenter for $\ell^2$ cost.

(d) **Our** EOT barycenter for $\ell^2$ cost (map from $\mathbb{P}_1$).

Figure 1: *2D twister example*: The true barycenter of 3 comets vs. the one computed by our solver with $\epsilon = 10^{-2}$. Two costs $c_k$ are considered: the twisted cost (1a, 1b) and $\ell^2$ (1c, 1d).

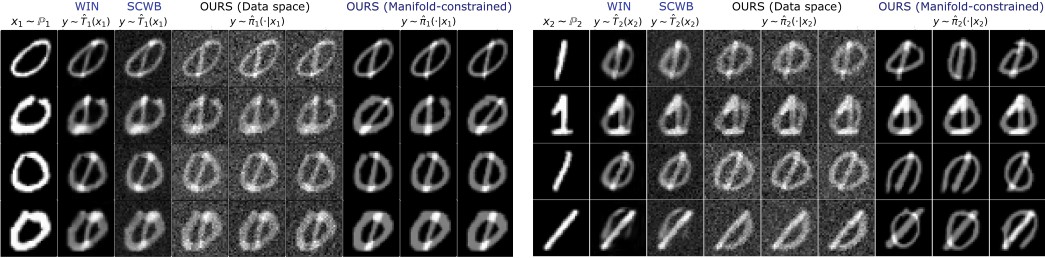

(a) Learned plans from $\mathbb{P}_1$ (zeros) to the barycenter.

(b) Learned plans from $\mathbb{P}_2$ (ones) to the barycenter.

Figure 2: Qualitative comparison of barycenters of MNIST 0/1 digit classes computed with barycenter solvers in the image space w.r.t. the pixel-wise $\ell^2$. Solvers SCWB and WIN only learn the unregularized barycenter ($\epsilon = 0$) directly in the data space. In turn, our solver learns the EOT barycenter in data space as well as it can learn EOT barycenter restricted to the StyleGAN manifold ($\epsilon = 10^{-2}$).

## 5.2 BARYCENTERS OF MNIST CLASSES 0 AND 1

A classic experiment considered in the continuous barycenter literature (Fan et al., 2021; Korotin et al., 2022a; Noble et al., 2023; Cohen et al., 2020) is averaging of distributions of MNIST 0/1 digits with weights $(\frac{1}{2}, \frac{1}{2})$ in the grayscale image space $\mathcal{X}_1 = \mathcal{X}_2 = \mathcal{Y} = [-1, 1]^{32 \times 32}$. The true unregularized ($\epsilon = 0$) $\ell^2$-barycenter images $y$ are direct pixel-wise averages $\frac{x_1 + x_2}{2}$ of pairs of images $x_1$ and $x_2$ coming from the $\ell^2$ OT plan between 0's ($\mathbb{P}_1$) and 1's ($\mathbb{P}_2$). In Fig. 2, we show the unregularized $\ell^2$ barycenter computed by (Fan et al., 2021, SCWB), (Korotin et al., 2022a, WIN).

**Data space EOT barycenter.** To begin with, we employ our solver to compute the $\epsilon$-regularized EOT $\ell^2$-barycenter directly in the image space $\mathcal{Y}$ for $\epsilon = 10^{-2}$. We emphasize that the true entropic barycenter slightly differs from the unregularized one. To be precise, it is expected that regularized barycenter images are close to the unregularized barycenter images but with additional noise. In Fig. 2, we see that our solver (data space) recovers the noisy barycenter images exactly as expected.

**Manifold-constrained EOT barycenter.** Averaging image distributions directly in the data space can be challenging. Our experiment below shows that if we *a priori* have some manifold $\mathcal{M}$ where we want the barycenter to be concentrated on, our solver can restrict the search space to it.

As discussed earlier, the support of the image-space unregularized $\ell^2$-barycenter is a certain *subset* of $\mathcal{M} = \{\frac{x_1 + x_2}{2} \mid x_1 \in \text{Supp}(\mathbb{P}_1), x_2 \in \text{Supp}(\mathbb{P}_1)\}$. To achieve this, we train a StyleGAN model $G : \mathcal{Z} \to \mathcal{Y}$ (Karras et al., 2019) with $\mathcal{Z} = \mathbb{R}^{512}$ to generate an approximate manifold $G(\mathcal{Z}) \approx \mathcal{M}$. Then, we use our solver with $\epsilon = 10^{-2}$ to search for the barycenter of 0/1 digit distributions on $\mathcal{X}_1, \mathcal{X}_2$ which lies in the latent space $\mathcal{Z}$ w.r.t. costs $c_k(x, z) \stackrel{\text{def}}{=} \frac{1}{2}\|x - G(z)\|^2$. This can be interpreted as learning the EOT $\ell^2$-barycenter in the ambient space but constrained to the StyleGAN-parameterized manifold $G(\mathcal{Z})$. In this case, the barycenter $\mathbb{Q}^*$ is the distribution of the latent variables $z$, which can be pushed to the manifold $G(\mathcal{Z}) \subset \mathcal{Y}$ via $G(z)$. The results are also in Fig. 2. There is no noise compared to the data-space EOT barycenter because of the manifold constraint.

We emphasize that the costs $c_k(x_k, z)$ used here are **general** (not $\ell^2$ cost!) because $G$ is a non-trivial StyleGAN generator. Hence, this experimental setup with the manifold-constrained barycenters is *not feasible for most other barycenter solvers* as they work exclusively with $\ell^2$.

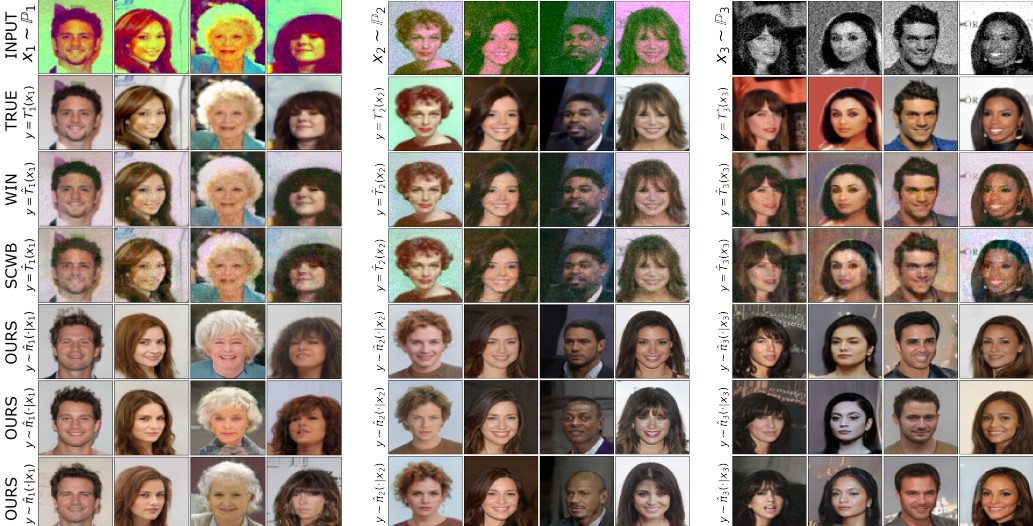

(a) Maps from $\mathbb{P}_1$ to the barycenter. (b) Maps from $\mathbb{P}_2$ to the barycenter. (c) Maps from $\mathbb{P}_3$ to the barycenter.

Figure 3: *Experiment on the Ave, celeba! barycenter dataset.* The plots compare the transported inputs $x_k \sim \mathbb{P}_k$ to the barycenter which are learned by various solvers. The true unregularized $\ell^2$ barycenter of $\mathbb{P}_1, \ldots, \mathbb{P}_2$ are the clean celebrity faces, see (Korotin et al., 2022a, §5).

## 5.3 EVALUATION ON THE AVE, CELEBA! DATASET

In (Korotin et al., 2022a), the authors developed a theoretically grounded methodology for finding probability distributions whose unregularized $\ell^2$ barycenter is known by construction. Based on the CelebA faces dataset (Liu et al., 2015), they constructed an Ave, celeba! dataset containing 3 degraded subsets of faces. The true $\ell^2$ barycenter w.r.t. the weights $(\frac{1}{4}, \frac{1}{2}, \frac{1}{4})$ is the distribution of Celeba faces itself. This dataset is used to test how well our approach recovers the barycenter.

We follow the EOT manifold-constrained setup as in §5.2 and train the StyleGAN on unperturbed celeba faces. This might sound a little bit unfair but our goal is to demonstrate the learned transport plan to the constrained barycenter rather than unconditional barycenter samples (recall the setup in §2.3). Hence, we learn the constrained EOT

| Solver | FID↓ of plans to the barycenter | | |
|---|---|---|---|
| | $k=1$ | $k=2$ | $k=3$ |
| (Fan et al., 2021) | 56.7 | 53.2 | 58.8 |
| (Korotin et al., 2022a) | 49.3 | 46.9 | 61.5 |
| **Ours** | **8.4** | **8.7** | **10.2** |

Table 2: FID scores of images mapped from inputs $\mathbb{P}_k$ to the barycenter.

barycenter with $\epsilon = 10^{-4}$. In Fig. 3, we present the results, depicting samples from the learned plans from each $\mathbb{P}_k$ to the barycenter. Overall, the map is qualitatively good, although sometimes failures in preserving the image content may occur. This is presumably due to MCMC inference getting stuck in local minima of the energy landscape. For comparison, we also show the results of the solvers by (Fan et al., 2021, SCWB), (Korotin et al., 2022a, WIN). Additionally, we report the FID score (Heusel et al., 2017) for images mapped to the barycenter in Table 2. Owing to the manifold-constrained setup, the FID score of our solver is significantly smaller.

## 6 DISCUSSION

**Potential impact.** In our work, we propose a novel approach for solving EOT barycenter problems which is applicable to *general OT costs*. From the practical viewpoint, we demonstrate the ability to restrict the sought-for barycenter to the *image manifold* by utilizing a pretrained generative model. Our findings may be applicable to a list of important real-world applications, see Appendix B.2. We believe that our large-scale barycenter solver will leverage industrial & socially-important problems.

**Limitations.** The limitations of our approach are mostly the same as those of EBMs. It is worth mentioning the usage of MCMC during the training/inference. The basic ULA algorithm which we use in §4.2 may poorly converge to the desired distribution $\mu_x^f$. In addition, MCMC sampling is time-consuming. We leave the search of more efficient sampling procedures for our solver, e.g., (Levy et al., 2017; Song et al., 2017; Habib & Barber, 2018; Neklyudov et al., 2020; Hoffman et al., 2019; Turitsyn et al., 2011; Lawson et al., 2019; Du et al., 2023), for future research.

## 7 REPRODUCIBILITY

We provide the source code for our solver and all the experiments in the supplementary material. The code is issued in a convenient form of `*.ipynb` notebooks.

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

# A  PROOFS

## A.1  AUXILIARY STATEMENTS

We start by showing some basic properties of the $C$-transform which will be used in the main proofs.

**Proposition 1** (Properties of the $C$-transform). *Let* $f_1, f_2 \colon \mathcal{Y} \to \mathbb{R}$ *be two measurable functions which are bounded from below. It holds that*

(i) **Monotonicity***:* $f_1 \leq f_2$ *implies* $f_1^C \geq f_2^C$;

(ii) **Constant additivity***:* $(f_1 + a)^C = f_1^C + a$ *for all* $a \in \mathbb{R}$;

(iii) **Concavity***:* $(\lambda f_1 + (1 - \lambda) f_2)^C \geq \lambda f_1^C + (1 - \lambda) f_2^C$ *for all* $\lambda \in [0, 1]$;

(iv) **Continuity***:* $f_1, f_2$ *bounded implies* $\sup_{x \in \mathcal{X}} |f_1^C(x) - f_2^C(x)| \leq \sup_{y \in \mathcal{Y}} |f_1(y) - f_2(y)|$.

*Proof of Proposition 1.* We recall the definition of the $C$-transform

$$f_1^C(x) = \inf_{\mu \in \mathcal{P}(\mathcal{Y})} \left\{ C(x, \mu) - \int_{\mathcal{Y}} f_1(y) \mathrm{d}\mu(y) \right\},$$

where $C(x, \mu) \stackrel{\text{def}}{=} \int_{\mathcal{Y}} c(x, y) \mathrm{d}\mu(y) - \epsilon H(\mu)$. Monotonicity (i) and constant additivity (ii) are immediate from the definition.

To see (iii), observe that the dependence of $\int_{\mathcal{Y}} f_1(y) \mathrm{d}\mu(y)$ on $f_1$ is linear. Thus, $f_1^C$ is the pointwise infimum of a family of linear functionals and thus concave.

Finally, to show (iv) we have by monotonicity of the integral that

$$\left| \int_{\mathcal{Y}} f_1(y) \mathrm{d}\mu(y) - \int_{\mathcal{Y}} f_2(y) \mathrm{d}\mu(y) \right| \leq \sup_{y \in \mathcal{Y}} |f_1(y) - f_2(y)| \tag{15}$$

for any $\mu \in \mathcal{P}(\mathcal{Y})$. For fixed $x \in \mathcal{X}$ we have

$$f_1^C(x) - f_2^C(x) = \inf_{\mu \in \mathcal{P}(\mathcal{Y})} \left[ C(x, \tilde{\mu}) - \int_{\mathcal{Y}} f_1(y) \mathrm{d}\tilde{\mu}(y) \right] - \inf_{\mu \in \mathcal{P}(\mathcal{Y})} \left[ C(x, \mu) - \int_{\mathcal{Y}} f_2(y) \mathrm{d}\mu(y) \right]$$

$$= \sup_{\mu \in \mathcal{P}(\mathcal{Y})} \inf_{\tilde{\mu} \in \mathcal{P}(\mathcal{Y})} \left[ C(x, \tilde{\mu}) - C(x, \mu) - \int_{\mathcal{Y}} f_1(y) \mathrm{d}\tilde{\mu}(y) + \int_{\mathcal{Y}} f_2(y) \mathrm{d}\mu(y) \right].$$

By setting $\tilde{\mu} = \mu$ we increase the value and obtain

$$f_1^C(x) - f_2^C(x) \leq \sup_{\mu \in \mathcal{P}(\mathcal{Y})} \int_{\mathcal{Y}} [f_2(y) - f_1(y)] \, \mathrm{d}\mu(y) \leq \sup_{y \in \mathcal{Y}} |f_1(y) - f_2(y)|, \tag{16}$$

where the last inequality follows from (15). For symmetry reasons, we can swap the roles of $f_1$ and $f_2$ in (16), which yields the claim. $\qquad\square$

## A.2  PROOF OF THEOREM 1

*Proof.* By substituting in (8) the primal EOT problems (2) with their dual counterparts (3), we obtain a dual formulation, which is the starting point of our analysis:

$$\mathcal{L}^* = \min_{\mathbb{Q} \in \mathcal{P}(\mathcal{Y})} \sup_{f_1, \dots, f_K \in \mathcal{C}(\mathcal{Y})} \underbrace{\sum_{k=1}^{K} \lambda_k \left\{ \int_{\mathcal{X}_k} f_k^{C_k}(x_k) \mathrm{d}\mathbb{P}_k(x_k) + \int_{\mathcal{Y}} f_k(y) \mathrm{d}\mathbb{Q}(y) \right\}}_{\stackrel{\text{def}}{=} \widetilde{\mathcal{L}}\left(\mathbb{Q}, \{f_k\}_{k=1}^K\right)}. \tag{17}$$

Here, we replaced $\inf$ with $\min$ because of the existence of the barycenter (§2.2). Moreover, we refer to the entire expression under the $\min \sup$ as a functional $\widetilde{\mathcal{L}} \colon \mathcal{P}(\mathcal{Y}) \times \mathcal{C}(\mathcal{Y})^K \to \mathbb{R}$. For brevity, we introduce, for $(f_1, \dots, f_K) \in \mathcal{C}(\mathcal{Y})^K$, the notation

$$\bar{f} \stackrel{\text{def}}{=} \sum_{k=1}^{K} \lambda_k f_k \quad \text{and} \quad M \stackrel{\text{def}}{=} \inf_{y \in \mathcal{Y}} \bar{f}(y) = \inf_{\mathbb{Q} \in \mathcal{P}(\mathcal{Y})} \int \bar{f}(y) \mathrm{d}\mathbb{Q}(y), \tag{18}$$

where the equality follows from two elementary observations that **(a)** $M \leq \int \bar{f}(y) \mathrm{d}\mathbb{Q}(y)$ for any $\mathbb{Q} \in \mathcal{P}(\mathcal{Y})$ and **(b)** $\bar{f}(y) = \int \bar{f}(y') \mathrm{d}\delta_y(y')$ where $\delta_y$ denotes a Dirac mass at $y \in \mathcal{Y}$.

On the one hand, $\mathcal{P}(\mathcal{Y})$ is compact w.r.t. the weak topology because $\mathcal{Y}$ is compact, and for fixed potentials $(f_1, \ldots, f_K) \in \mathcal{P}(\mathcal{Y})^K$ we have that $\widetilde{\mathcal{L}}(\cdot, (f_k)_{k=1}^K)$ is continuous and linear. In particular, $\widetilde{\mathcal{L}}(\cdot, (f_k)_{k=1}^K)$ is convex and l.s.c. On the other hand, for a fixed $\mathbb{Q}$, the functional $\widetilde{\mathcal{L}}(\mathbb{Q}, \cdot)$ is concave by (iii) in Proposition 1. These observations allow us to apply Sion's minimax theorem (Sion, 1958, Theorem 3.4) to swap $\min$ and $\inf$ in (17) and obtain using (18)

$$
\begin{aligned}
\mathcal{L}^* &= \sup_{f_1, \ldots, f_K \in \mathcal{C}(\mathcal{Y})} \min_{\mathbb{Q} \in \mathcal{P}(\mathcal{Y})} \sum_{k=1}^K \lambda_k \left\{ \int_{\mathcal{X}_k} f_k^{C_k}(x_k) \mathrm{d}\mathbb{P}_k(x_k) + \int_{\mathcal{X}} f_k(y) \mathrm{d}\mathbb{Q}(y) \right\} \\
&= \sup_{f_1, \ldots, f_K \in \mathcal{C}(\mathcal{Y})} \left\{ \sum_{k=1}^K \lambda_k \int_{\mathcal{X}_k} f_k^{C_k}(x_k) \mathrm{d}\mathbb{P}_k(x_k) + \min_{\mathbb{Q} \in \mathcal{P}(\mathcal{Y})} \int_{\mathcal{X}} \bar{f}(y) \mathrm{d}\mathbb{Q}(y) \right\} \\
&= \sup_{f_1, \ldots, f_K \in \mathcal{C}(\mathcal{Y})} \underbrace{\left\{ \sum_{k=1}^K \lambda_k \int_{\mathcal{X}_k} f_k^{C_k}(x_k) \mathrm{d}\mathbb{P}_k(x_k) + \min_{y \in \mathcal{Y}} \bar{f}(y) \right\}}_{\overset{\text{def}}{=} \widetilde{\mathcal{L}}(f_1, \ldots, f_K)} .
\end{aligned} \tag{19}
$$

Next, we show that the $\sup$ in (19) can be restricted to tuplets satisfying the congruence condition $\sum_{k=1}^K \lambda_k f_k = 0$. It remains to show that for every tuplet $(f_1, \ldots, f_K) \in \mathcal{C}(\mathcal{Y})^K$ there exists a *congruent* tuplet $(\tilde{f}_1, \ldots, \tilde{f}_K) \in \mathcal{C}(\mathcal{Y})^K$ such that $\widetilde{\mathcal{L}}(\tilde{f}_1, \ldots, \tilde{f}_K) \geq \widetilde{\mathcal{L}}(f_1, \ldots, f_K)$.

To this end, fix $(f_1, \ldots, f_K)$ and define the congruent tuplet

$$
(\tilde{f}_1, \ldots, \tilde{f}_K) \overset{\text{def}}{=} \left( f_1, \ldots, f_{K-1}, f_K - \frac{\bar{f}}{\lambda_K} \right). \tag{20}
$$

We find $\tilde{M} \overset{\text{def}}{=} \inf_{y \in \mathcal{Y}} \sum_{k=1}^K \lambda_k \tilde{f}_k = 0$ by the congruence and derive

$$
\begin{aligned}
\widetilde{\mathcal{L}}(\tilde{f}_1, \ldots, \tilde{f}_K) - \widetilde{\mathcal{L}}(f_1, \ldots, f_K) &= \lambda_K \int_{\mathcal{X}_K} \left[ \tilde{f}_K^{C_K}(x_K) - f_K^{C_K}(x_K) \right] \mathrm{d}\mathbb{P}_K(x_K) - M \\
&\geq \lambda_K \int_{\mathcal{X}_K} \left[ \left( f_K - \frac{M}{\lambda_K} \right)^{C_K}(x_K) - f_K^{C_K}(x_K) \right] \mathrm{d}\mathbb{P}(x_K) - M \\
&= \lambda_K \int_{\mathcal{X}_K} \frac{M}{\lambda_K} \mathrm{d}\mathbb{P}(x_K) - M = 0,
\end{aligned}
$$

where the first inequality follows from $\tilde{f}_K = f_K - \frac{\bar{f}}{\lambda_K} \leq f_K - \frac{M}{\lambda_K}$ combined with monotonicity of the $C$-transform, see (i) in Proposition 1. The second to last equality follow from constant additivity, see (ii) in Proposition 1.

In summary, we obtain

$$
\mathcal{L}^* = \sup_{\substack{f_1, \ldots, f_k \in \mathcal{C}(\mathcal{Y}) \\ \sum_{k=1}^K f_k = 0}} \widetilde{\mathcal{L}}(f_1, \ldots, f_K). \tag{21}
$$

Finally, observe that for congruent $(f_1, \ldots, f_K)$ we have $\widetilde{\mathcal{L}}(f_1, \ldots, f_K) = \mathcal{L}(f_1, \ldots, f_K)$. Hence, we can replace $\widetilde{\mathcal{L}}$ by $\mathcal{L}$ in (21), which yields (9). $\qquad \square$

### A.3 PROOF OF THEOREM 2

*Proof.* Write $\mathbb{Q}^*$ for the barycenter and $\pi_k^*$ for the optimizer of $\mathrm{EOT}_{c_k, \epsilon}(\mathbb{P}_k, \mathbb{Q}^*)$. Consider congruent potentials $f_1, \ldots, f_K \in \mathcal{C}(\mathcal{Y})$ and define the probability distribution

$$
\mathrm{d}\pi^{f_k}(x_k, y) \overset{\text{def}}{=} \mathrm{d}\mu_{x_k}^{f_k}(y) \, \mathrm{d}\mathbb{P}_k(x_k),
$$

where

$$\frac{\mathrm{d}\mu_{x_k}^{f_k}(y)}{\mathrm{d}y} \stackrel{\text{def}}{=} \frac{1}{Z_{c_k}(f_k, x_k)} \exp\left(\frac{f_k(y) - c_k(x_k, y)}{\epsilon}\right),$$

$$Z_{c_k}(f_k, x_k) \stackrel{\text{def}}{=} \log\left(\int_{\mathcal{Y}} e^{\frac{f_k(y) - c_k(x_k, y)}{\epsilon}} \mathrm{d}y\right).$$

Then we have by (6)

$$\mathrm{EOT}_{c_k, \epsilon}(\mathbb{P}_k, \mathbb{Q}^*) - \left(\int_{\mathcal{X}_k} f^{C_k}(x_k)\mathrm{d}\mathbb{P}(x_k) + \int_{\mathcal{Y}} f(y)\mathrm{d}\mathbb{Q}^*(y)\right) = \epsilon\mathrm{KL}\left(\pi_k^*\|\pi^{f_k}\right). \qquad (22)$$

Multiplying (22) by $\lambda_k$ and summing over $k$ yields

$$\epsilon\sum_{k=1}^{K}\lambda_k\mathrm{KL}\left(\pi_k^*\|\pi^{f_k}\right) = \sum_{k=1}^{K}\lambda_k\left\{\mathrm{EOT}_{c_k, \epsilon}(\mathbb{P}_k, \mathbb{Q}^*) - \int_{\mathcal{X}_k} f_k^{C_k}(x_k)\mathrm{d}\mathbb{P}_k(x_k)\right\} - \int_{\mathcal{Y}}\underbrace{\sum_{k=1}^{K}\lambda_k f(y)}_{=0}\mathrm{d}\mathbb{Q}^*(y)$$

$$= \mathcal{L}^* - \mathcal{L}(f_1, \ldots, f_k),$$

where the last equality follows by congruence, i.e., $\sum_{k=1}^{K}\lambda_k f_k \equiv 0$.

The remaining inequality in (10) is a consequence of the data processing inequality for $f$-divergences which we invoke here to get

$$\mathrm{KL}\left(\pi_k^*\|\pi^{f_k}\right) \geq \mathrm{KL}\left(\mathbb{Q}^*\|\mathbb{Q}^{f_k}\right),$$

where $\mathbb{Q}^*$ and $\mathbb{Q}^{f_k}$ are the second marginals of $\pi_k^*$ and $\pi^{f_k}$, respectively. $\qquad \square$

### A.4    PROOF OF THEOREM 3

*Proof.* The desired equation (12) could be derived exactly the same way as it is done in (Mokrov et al., 2023, Theorem 3). $\qquad \square$

Before proving Theorem 4, we recall the required basic concepts of statistical learning theory (Shalev-Shwartz & Ben-David, 2014, §26). Consider some class $\mathcal{H}$ of functions $h : \mathcal{X} \to \mathbb{R}$ and a distribution $\mu$ on $\mathcal{X}$. Let $X = \{x^1, \ldots, x^N\}$ be a sample of $N$ points in $\mathcal{X}$.

The **representativeness** of the sample $X$ w.r.t. the class $\mathcal{H}$ and the distribution $\mathbb{P}$ is defined by

$$\mathrm{Rep}_X(\mathcal{H}, \mu) \stackrel{\text{def}}{=} \sup_{h \in \mathcal{H}}\left[\int_{\mathcal{X}} h(x)\mathrm{d}\mu(x) - \frac{1}{N}\sum_{n=1}^{N} h(x^n)\right]. \qquad (23)$$

The **Rademacher complexity** of the class $\mathcal{H}$ w.r.t. the distribution $\mathbb{P}$ and sample size $N$ is given by

$$\mathcal{R}_N(\mathcal{H}, \mu) \stackrel{\text{def}}{=} \frac{1}{N}\mathbb{E}\left\{\sup_{h \in \mathcal{H}}\sum_{n=1}^{N} h(x^n)\sigma_n\right\}, \qquad (24)$$

where $\{x^n\}_{n=1}^{N} \sim \mu$ are mutually independent, $\{\sigma^n\}_{n=1}^{N}$ are mutually independent Rademacher random variables, i.e., $\mathrm{Prob}(\sigma^n = 1) = \mathrm{Prob}(\sigma^n = -1) = 0.5$, and the expectation is taken with respect to both $\{x_n\}_{n=1}^{N}, \{\sigma_n\}_{n=1}^{N}$. The well-celebrated relation between (24) and (23) is

$$\mathbb{E}\mathrm{Rep}_X(\mathcal{H}, \mu) \leq 2 \cdot \mathcal{R}_N(\mathcal{F}, \mu), \qquad (25)$$

where the expectation is taken w.r.t. random i.i.d. sample $X \sim \mu$ of size $N$.

## A.5 PROOF OF THEOREM 4

*Proof.*

$$\epsilon \sum_{k=1}^{K} \lambda_k \text{KL}\left(\pi_k^* \| \pi^{\widehat{f}_k}\right) = \mathcal{L}^* - \mathcal{L}(\widehat{\mathbf{f}}) = \underbrace{\left[\mathcal{L}^* - \max_{\mathbf{f} \in \overline{\mathcal{F}}} \mathcal{L}(\mathbf{f})\right]}_{\text{Approximation error}} + \underbrace{\left[\max_{\mathbf{f} \in \overline{\mathcal{F}}} \mathcal{L}(\mathbf{f}) - \mathcal{L}(\widehat{\mathbf{f}})\right]}_{\text{Estimation error}}. \tag{26}$$

Let $\bar{\mathbf{f}}$ be an $\arg\max_{\mathbf{f} \in \overline{\mathcal{F}}} \mathcal{L}(\mathbf{f})$. We conduct the standard analysis for the estimation error in (26):

$$\begin{aligned}
\max_{\mathbf{f} \in \overline{\mathcal{F}}} \mathcal{L}(\mathbf{f}) - \mathcal{L}(\widehat{\mathbf{f}}) &= \mathcal{L}(\bar{\mathbf{f}}) - \mathcal{L}(\widehat{\mathbf{f}}) \\
&= \left[\mathcal{L}(\bar{\mathbf{f}}) - \widehat{\mathcal{L}}(\bar{\mathbf{f}})\right] + \underbrace{\left[\widehat{\mathcal{L}}(\bar{\mathbf{f}}) - \widehat{\mathcal{L}}(\widehat{\mathbf{f}})\right]}_{\leq 0} + \left[\widehat{\mathcal{L}}(\widehat{\mathbf{f}}) - \mathcal{L}(\widehat{\mathbf{f}})\right] \\
&\leq \sup_{\mathbf{f} \in \overline{\mathcal{F}}} \left[\mathcal{L}(\mathbf{f}) - \widehat{\mathcal{L}}(\mathbf{f})\right] + 0 + \sup_{\mathbf{f} \in \overline{\mathcal{F}}} \left[\widehat{\mathcal{L}}(\mathbf{f}) - \mathcal{L}(\mathbf{f})\right] \tag{27} \\
&\leq 2 \sup_{\mathbf{f} \in \overline{\mathcal{F}}} \left[\mathcal{L}(\mathbf{f}) - \widehat{\mathcal{L}}(\mathbf{f})\right] \\
&\leq 2 \sum_{k=1}^{K} \lambda_k \sup_{f_k \in \mathcal{F}_k} \left[\int_{\mathcal{X}_k} f_k^{C_k}(x_k) \mathrm{d}\mathbb{P}_k(x_k) - \frac{1}{N_k} \sum_{n=1}^{N_k} f_k^{C_k}(x_k^n)\right] \\
&= 2 \sum_{k=1}^{K} \lambda_k \text{Rep}_{X_k}(\mathcal{F}_k^{C_k}, \mathbb{P}_n). \tag{28}
\end{aligned}$$

To upper bound the central term in transition to line (27), we use maximality of $\widehat{\mathbf{f}}$, that is, $\widehat{\mathcal{L}}(\widehat{\mathbf{f}}) = \max_{\mathbf{f} \in \overline{\mathcal{F}}} \widehat{\mathcal{L}}(\mathbf{f}) \geq \widehat{\mathcal{L}}(\bar{\mathbf{f}})$. We plug (28) into (26), take the expectation and obtain

$$\begin{aligned}
\epsilon \mathbb{E} \sum_{k=1}^{K} \lambda_k \text{KL}\left(\pi_k^* \| \pi^{\widehat{f}_k}\right) &\leq \left[\mathcal{L}^* - \max_{\mathbf{f} \in \overline{\mathcal{F}}} \mathcal{L}(\mathbf{f})\right] + 2\mathbb{E} \sum_{k=1}^{K} \lambda_k \text{Rep}_{X_k}(\mathcal{F}_k^{C_k}, \mathbb{P}_n) \\
&\leq \left[\mathcal{L}^* - \max_{\mathbf{f} \in \overline{\mathcal{F}}} \mathcal{L}(\mathbf{f})\right] + 4\mathbb{E} \sum_{k=1}^{K} \lambda_k \mathcal{R}_{N_k}(\mathcal{F}^{C_k}, \mathbb{P}_k),
\end{aligned}$$

where the last inequality is the usual Rademacher bound (25). $\qquad\square$

## A.6 PROOF OF THEOREM 5

*Proof.*

$$\mathcal{L}^* - \mathcal{L}(f_1', \ldots, f_K') < \frac{\delta}{2}. \tag{29}$$

These are continuous functions defined on a compact subset of $\mathbb{R}^{D_k}$. From the general approximation theory it follows that for every $\delta_1, \ldots, \delta_K > 0$ there exist neural networks $g_k : \mathbb{R}^{D_k} \to \mathbb{R}$ $(k \in \overline{K})$ which satisfy $\|g_k - f_k'\|_\infty \overset{\text{def}}{=} \sup_{y \in \mathcal{Y}} |g_k(y) - f_k'(y)| < \delta_k$. For this, we may use (Kidger & Lyons, 2020, Theorem 3.2) or any other known neural universal approximation theorem. Pick $\delta_k = \frac{\delta}{4}$ for all $k \in \overline{K}$ and suitable neural networks $g_1, \ldots, g_K$. Next, we define the congruent sums of neural networks $f_k \overset{\text{def}}{=} g_k - \sum_{k=1}^{K} \lambda_k g_k$. We derive

$$\|\sum_{k=1}^{K} \lambda_k g_k\|_\infty = \|\sum_{k=1}^{K} \lambda_k g_k - \underbrace{\sum_{k=1}^{K} \lambda_k f_k'}_{=0}\|_\infty \leq \sum_{k=1}^{K} \lambda_k \|g_k - f_k'\|_\infty < \sum_{k=1}^{K} \lambda_k \delta_k = \frac{\delta}{4}. \tag{30}$$

Using (30) we obtain for fixed $k \in \overline{K}$

$$\|f_k' - f_k\|_\infty = \|f_k' - g_k + \sum_{k'=1}^{K} \lambda_{k'} g_{k'}\|_\infty \leq \underbrace{\|f_k' - g_k\|_\infty}_{<\frac{\delta}{4}} + \underbrace{\|\sum_{k'=1}^{K} \lambda_{k'} g_{k'}\|_\infty}_{<\frac{\delta}{4}} < \frac{\delta}{2}. \quad (31)$$

By (iv) in Proposition 1 together with (31) we find

$$\|f_k^{C_k} - (f_k')^{C_k}\|_\infty \leq \|f_k - f_k'\|_\infty < \frac{\delta}{2}. \quad (32)$$

Now we use (32) to derive

$$|\mathcal{L}(f_1, \ldots, f_K) - \mathcal{L}(f_1', \ldots, f_K')| \leq \sum_{k=1}^{K} \lambda_k \left| \int_{\mathcal{X}_k} f_k^{C_k}(x_k) d\mathbb{P}_k(x_k) - \int_{\mathcal{X}_k} (f_k')^{C_k}(x_k) d\mathbb{P}_k(x_k) \right|$$

$$\leq \sum_{k=1}^{K} \lambda_k \int_{\mathcal{X}_k} |f_k^{C_k}(x_k) - (f_k')^{C_k}(x_k)| d\mathbb{P}_k(x_k)$$

$$\leq \sum_{k=1}^{K} \lambda_k \|f_k^{C_k} - (f_k')^{C_k}\|_\infty < \underbrace{(\sum_{k=1}^{K} \lambda_k)}_{=1} \frac{\delta}{2} = \frac{\delta}{2}. \quad (33)$$

Next we combine (29) with (33) to get

$$\mathcal{L}^* - \mathcal{L}(f_1, \ldots, f_K) \leq \underbrace{[\mathcal{L}^* - \mathcal{L}(f_1', \ldots, f_K')]}_{<\delta/2} + \underbrace{|\mathcal{L}(f_1, \ldots, f_K) - \mathcal{L}(f_1', \ldots, f_K')|}_{<\delta/2} < \delta. \quad (34)$$

By using (34) together with Theorem 2 we obtain

$$\epsilon \sum_{k=1}^{K} \lambda_k \text{KL}\left(\pi_k^* \| \pi^{f_k}\right) = \mathcal{L}^* - \mathcal{L}(f_1, \ldots, f_K) < \delta$$

which finishes the proof. $\qquad\square$

# B    EXTENDED DISCUSSIONS

## B.1    EXTENDED DISCUSSION OF RELATED WORKS

**Discrete OT-based solvers** provide solutions to OT-related problems between discrete distributions. A comprehensive overview can be found in (Peyré et al., 2019). The discrete OT methods for EOT barycenter estimation are (Cuturi & Doucet, 2014; Solomon et al., 2015; Benamou et al., 2015; Cuturi & Peyré, 2016; Cuturi & Peyré, 2018; Dvurechenskii et al., 2018; Krawtschenko et al., 2020), (Le et al., 2021). In spite of sound theoretical foundations and established convergence guarantees (Kroshnin et al., 2019), these approaches can not be directly adapted to our learning setup, see §2.3.

**Continuous OT solvers.** Beside the continuous EOT solvers discussed in §3, there exist a variety of neural OT solver for the non-entropic (unregularized, $\epsilon = 0$) case. For example, solvers such as (Henry-Labordere, 2019; Makkuva et al., 2020; Korotin et al., 2021a;b; 2022b; Fan et al., 2021; Gazdieva et al., 2023; Rout et al., 2021; Amos, 2022), are based on optimizing the dual form, similar to our (3), with neural networks. We mention these methods because they serve as the basis for certain continuous unregularized barycenter solvers. For example, ideas of (Korotin et al., 2022a) are employed in the barycenter method presented in (Korotin et al., 2021c); the solver from (Makkuva et al., 2020) is applied in (Fan et al., 2021); max-min solver introduced in (Korotin et al., 2021b) is used in (Korotin et al., 2022a). It is also worth noting that there exist several neural solvers that cater to more general OT problem formulations (Korotin et al., 2023b;a; Asadulaev et al., 2022). These can even be adapted to the EOT case (Gushchin et al., 2023b) but require substantial technical effort and the usage of restrictive neural architectures.

**Other related works.** Another relevant work is (Simon & Aberdam, 2020), where the authors study the barycenter problem and restrict the search space to a manifold produced by a GAN. This idea is

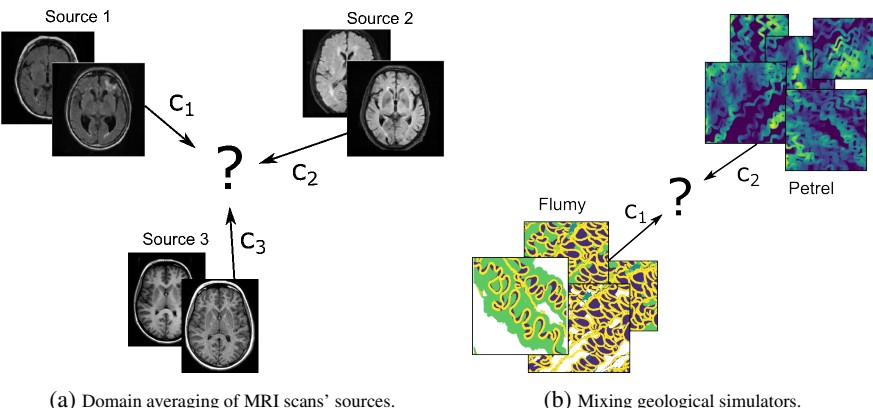

(a) Domain averaging of MRI scans' sources.        (b) Mixing geological simulators.

Figure 4: A schematical presentation of potential applications of barycenter solvers.

also utilized in §5.2 and §5.3, but their overall setting drastically differs from our setup and actually is not applicable. We search for a barycenter of $K$ *high-dimensional image distributions* represented by their random samples (datasets). In contrast, they consider $K$ images, represent each *image as a 2D distribution* via its intensity histogram and search for a **single image** on the GAN manifold whose density is the barycenter of the input images. To compute the barycenter, they use discrete OT solver. In summary, neither our barycenter solver is intended to be used in their setup, nor their method is targeted to solve the problems considered in our paper.

### B.2    EXTENDED DISCUSSION OF POTENTIAL APPLICATIONS

It is not a secret that despite considerable efforts in developing continuous barycenter solvers (Li et al., 2020; Korotin et al., 2021c; 2022a; Fan et al., 2021; Noble et al., 2023; Chi et al., 2023), these solvers have not found yet a real working practical application. The reasons for this are two-fold:

1. Existing continuous barycenter solvers (Table 1) are yet not scalable enough and/or work exclusively with the quadratic cost ($\ell^2$), which might be not sufficient for the practical needs.
2. Potential applications of barycenter solvers are too technical and, unfortunately, require substantial efforts (challenging and costly data collection, non-trivial design of task-specific cost functions, unclear performance metrics, etc.) to be implemented in practice.

Despite these challenges, there exist rather inspiring practical problem formulations where the continuous barycenter solvers may potentially shine and we name a few below. These potential applications motivate the research in the area. More generally, we hope that our developed solver could be a step towards applying continuous barycenters to practical tasks that benefit humanity.

**1. Solving domain shift problems in medicine (Fig. 4a).** In medicine, it is common that the data is collected from multiple sources (laboratories, clinics) and using different equipment from various vendors, each with varying technical characteristics (Guan & Liu, 2021; Kushol et al., 2023; Kondrateva et al., 2021; Stacke et al., 2020; Yan et al., 2019). Moreover, the data coming from each source may be of limited size. These issues complicate building robust and reliable machine learning models by using such datasets, e.g., learning MRI segmentation models to assist doctors.

A potential way to overcome the above-mentioned limitations is to find a common representation of the data coming from multiple sources. This representation would require translation maps that can transform the new (previously unseen) data from each of the sources to this shared representation. This formulation closely aligns with the continuous barycenter learning setup (§2.3) studied in our paper. In this context, the barycenter could play the role of the shared representation.

To apply barycenters effectively to such domain averaging problems, two crucial ingredients are likely required: appropriate cost functions $c_k$ and a suitable data manifold $\mathcal{M}$ in which to search for the barycenters. The design of the cost itself may be a challenge requiring certain domain-specific knowledge that necessitates involving experts in the field. Meanwhile, the manifold constraint is required to avoid practically meaningless barycenters such as those considered in §5.2. Nowadays,

with the rapid growth of the field of generative models, it is reasonable to expect that soon the new large models targeted for medical data may appear, analogously to DALL-E (Ramesh et al., 2022), StableDiffusion (Rombach et al., 2022) or StyleGAN-T (Sauer et al., 2023) for general image synthesis. These future models could potentially parameterize the medical data manifolds of interest, opening new possibilities for medical data analysis.

**2. Mixing geological simulators (Fig. 4b).** In geological modeling, variuos simulators exist to model different aspects of underground deposits. Sometimes one needs to build a generic tool which can take into account several desired geological factors which are successfully modeled by separate simulators.

**Flumy**[1] is a process-based simulator that uses hydraulic theory (Ikeda et al., 1981) to model specific channel depositional processes returning a detailed three-dimensional geomodel informed with deposit lithotype, age and grain size. However, its result is a 3D segmentation field of facies (rock types) and it does not produce the real valued porosity field needed for hydrodynamical modeling.

**Petrel**[2] software is the other popular simulator in the oil and gas industry. It is able to model complex real-valued geological maps such as the distribution of porosity. The produced porosity fields may not be realistic enough due to paying limited attention to the geological formation physics.

Both Flumy and Petrel simulators contain some level of stochasticity and are hard to use in conjunction. Even when conditioned on common prior information about the deposit, they may produce maps of facies and permeability which do not meaningfully correspond to each other. This limitation provides potential *prospects for barycenter solvers* which could be used to *get the best from both simulators* by mixing the distributions produced by each of them.

From our personal discussions with the experts in the field of geology, such task formulations are of considerable interest both for scientific community as well as industry. Applying our barycenter solver in this context is a challenge for future research. We acknowledge that this would also require overcoming considerable technical and domain-specific issues, including the data collection and the choice of costs $c_k$.

## C   EXPERIMENTAL DETAILS

The hyper-parameters of our solver are summarized in Table 3. Working with the manifold-constraint setup, we parameterize each $g_{\theta_k}(z)$ in our sover as $h_{\theta_k} \circ G(z)$, where $G$ is a pre-trained (frozen) StyleGAN and $h_{\theta_k}$ is a neural network with the ResNet architecture. We empirically found that this strategy provides better results than a direct MLP parameterization for the function $g_{\theta_k}(z)$.

| Experiment | $D$ | $K$ | $\epsilon$ | $\lambda_1$ | $\lambda_2$ | $\lambda_3$ | $f_{\theta,k}$ | $lr_{g_{\theta,k}}$ | $iter$ | $\sqrt{\eta}$ | $L$ | $S$ |
|---|---|---|---|---|---|---|---|---|---|---|---|---|
| Toy 2D | 2 | 3 | $10^{-2}$ | 1/3 | 1/3 | 1/3 | MLP | $10^{-2}$ | 200 | 1.0 | 300 | 256 |
| MNIST 0/1 | 1024 | 2 | $10^{-2}$ | 0.5 | 0.5 | - | ResNet | $10^{-4}$ | 1000 | 0.1 | 500 | 32 |
| MNIST 0/1 | 512 | 2 | $10^{-2}$ | 0.5 | 0.5 | - | ResNet | $10^{-4}$ | 1000 | 0.1 | 250 | 32 |
| Ave, celeba! | 512 | 3 | $10^{-4}$ | 0.25 | 0.5 | 0.25 | ResNet | $10^{-4}$ | 1000 | 0.1 | 250 | 128 |
| Gaussians | 2-64 | 3 | $10^{-2}, 1$ | 0.25 | 0.25 | 0.5 | MLP | $10^{-3}$ | 50000 | 0.1 | 700 | 1024 |

Table 3: Hyperparameters that we use in the experiments with our algorithm 1.

To train the StyleGAN for MNSIT01 & Ave, celeba! experiments, we employ the official code from

```
https://github.com/NVlabs/stylegan2-ada-pytorch
```

### C.1   BARYCENTERS OF 2D DISTRIBUTIONS

In this subsection we provide a mathematical derivation that the true unregularized barycenter of the distributions $\mathbb{P}_1, \mathbb{P}_2, \mathbb{P}_3$ in Fig. 1a coincides with a Gaussian.

---
[1]https://flumy.minesparis.psl.eu
[2]https://www.software.slb.com/products/petrel

We begin with a rather general observation. Consider $\mathcal{X}_k = \mathcal{Y} = \mathbb{R}^D$ ($k \in \overline{K}$) and let $\text{OT}_c \overset{\text{def}}{=}$ $\text{EOT}_{c,0}$ denote the unregularized OT problem ($\epsilon = 0$) for a given continuous cost function $c$. Let $u : \mathbb{R}^D \to \mathbb{R}^D$ be a measurable bijection and consider $\mathbb{P}'_k \in \mathcal{P}(\mathbb{R}^D)$ for $k \in \overline{K}$. By using the change of variables formula, we have for all $\mathbb{Q}' \in \mathcal{P}(\mathbb{R}^D)$ that

$$\text{OT}_{c \circ (u \times u)}(u_\#^{-1}(\mathbb{P}'), u_\#^{-1}(\mathbb{Q}')) = \text{OT}_c(\mathbb{P}', \mathbb{Q}'), \tag{35}$$

where $\#$ denotes the pushforward operator of distributions and $[c \circ (u \times u)](x, y) = c\big(u(x), u(y)\big)$. Note that by (35) the barycenter of $\mathbb{P}'_1, \ldots, \mathbb{P}'_K$ for the unregularized problem with cost $c$ coincides with the result of applying the pushforward operator $u_\#^{-1}$ to the barycenter of the very same problem but with cost $c \circ (u \times u)$.

Next, we fix $u$ to be the twister map (§5.1). In Fig. 1a we plot the distributions $\mathbb{P}_1 \overset{\text{def}}{=}$ $u_\#^{-1}\mathbb{P}'_1, u_\#^{-1}\mathbb{P}'_1, u_\#^{-1}\mathbb{P}'_3$ which are obtained from Gaussian distributions $\mathbb{P}'_1 = \mathcal{N}\big((0, 4), I_2\big), \mathbb{P}'_2 = \mathcal{N}\big((-2, 2\sqrt{3}), I_2\big), \mathbb{P}'_3 = \mathcal{N}\big((2, 2\sqrt{3}), I_2\big)$ by the pushforward. Here $I_2$ is the 2-dimensional identity matrix. For the unregularized $\ell^2$ barycenter problem, the barycenter of such shifted Gaussians can be derived analytically (Álvarez-Esteban et al., 2016). The solution coincides with a zero-centered standard Gaussian $\mathbb{Q}' = \mathcal{N}\big(0, I_2\big)$. Hence, the barycenter of $\mathbb{P}_1, \ldots, \mathbb{P}_K$ w.r.t. the cost $\ell^2 \circ (u \times u)$ is given by $\mathbb{Q}^* = u_\#^{-1}\mathbb{Q}'$. From the particular choice of $u$ it is not hard to see that $\mathbb{Q}^* = \mathbb{Q}' = \mathcal{N}\big(0, I_2\big)$ as well.

## C.2 BARYCENTERS OF MNIST CLASSES

**Additional qualitative examples** of our solver's results are given in Figure 5.

**Details of the baseline solvers.** For the solver by (Fan et al., 2021, SCWB), we run their publicly available code from the official repository

```
https://github.com/sbyebss/Scalable-Wasserstein-Barycenter
```

The authors do no provide checkpoints, and we train their barycenter model from scratch. In turn, for the solver by (Korotin et al., 2022a, WIN), we also employ the publicly available code

```
https://github.com/iamalexkorotin/WassersteinIterativeNetworks
```

Here we do not train their models but just use the checkpoint available in their repo.

## C.3 BARYCENTERS OF THE AVE,CELEBA! DATASET

**Additional qualitative examples** of our solver's results are given in Figure 6.

**Details of the baseline solvers.** For the (Korotin et al., 2022a, WIN) solver, we use their pre-trained checkpoints provided by the authors in the above-mentioned repository. Note that the authors of (Fan et al., 2021, SCWB) do not consider such a high-dimensional setup with RGB images. Hence, to implement their approach in this setting, we follow (Korotin et al., 2022a, Appendix B.4).

## C.4 BARYCENTERS OF GAUSSIAN DISTRIBUTIONS

We note that there exist many ways to incorporate the entropic regularization for barycenters (Chizat, 2023, Table 1); these problems do not coincide and yield **different** solutions. For some of them, the ground-truth solutions are known for specific cases such as the Gaussian case. For example, (Mallasto et al., 2022, Theorem 3) examines barycenters for OT regularized with KL divergence. They consider the task

$$\inf_{\mathbb{Q} \in \mathcal{P}(\mathcal{Y})} \sum_{k=1}^K \lambda_k \big( \int_{\mathcal{X} \times \mathcal{Y}} \frac{\|x - y\|^2}{2} d\pi_k(x, y) + \epsilon \text{KL}(\pi_k \| \mathbb{P}_k \times \mathbb{Q})\big) =$$

$$\inf_{\mathbb{Q} \in \mathcal{P}(\mathcal{Y})} \sum_{k=1}^K \lambda_k \big( \int_{\mathcal{X} \times \mathcal{Y}} \frac{\|x - y\|^2}{2} d\pi_k(x, y) - \epsilon \int_{\mathcal{X}} H(\pi_k(\cdot | x)) d\mathbb{P}_k(x) + \epsilon H(\mathbb{Q})\big) =$$

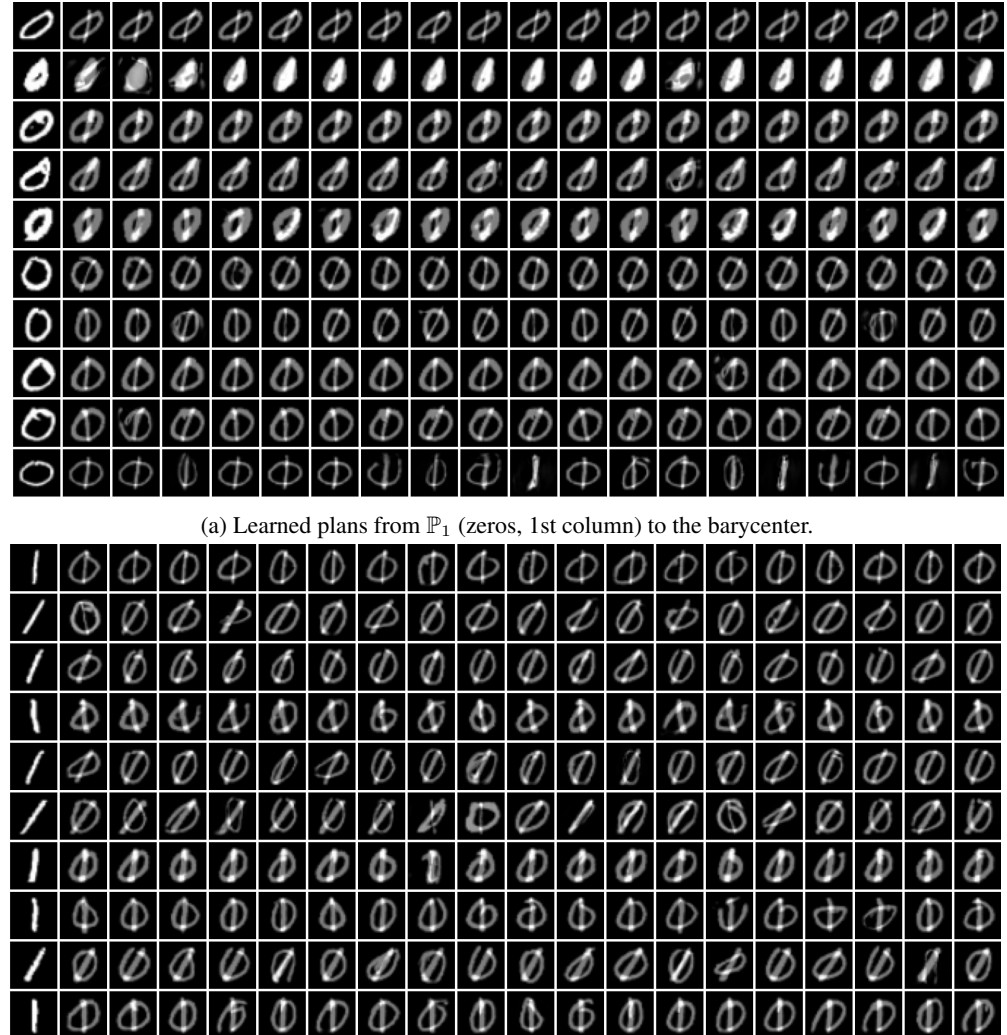

(a) Learned plans from $\mathbb{P}_1$ (zeros, 1st column) to the barycenter.

(b) Learned plans from $\mathbb{P}_2$ (ones, first column) to the barycenter.

Figure 5: *Experiment with averaging MNIST 0/1 digit classes.* The plot shows additional examples of samples transported with **our** solver to the barycenter.

$$\inf_{\mathbb{Q} \in \mathcal{P}(\mathcal{Y})} \left\{ \underbrace{\sum_{k=1}^{K} \lambda_k \text{EOT}^{(2)}_{\ell^2, \epsilon}(\mathbb{P}_k, \mathbb{Q})}_{= \text{ our objective inside (7)}} + \epsilon H(\mathbb{Q}) \right\}.$$

This problem differs from our objective (7) with $c(x, y) = \frac{1}{2}\|x - y\|^2$ by the non-constant $\mathbb{Q}$-**dependent** term $\epsilon H(\mathbb{Q})$; this problem yields a different solution. The difference of other mentioned approaches can be shown in the same way. In particular, (Le et al., 2022) tackles the barycenter for inner product Gromov-Wasserstein problem with entropic regularization which is not relevant for us. To our knowledge, the Gaussian ground-truth solution for our problem setup (7) is not yet known, although some of its properties seem to be established (del Barrio & Loubes, 2020).

Still when $\epsilon \approx 0$, our entropy-regularized barycenter is expected to be close to the unregularized one ($\epsilon = 0$). In the Gaussian case, it is known that the unregularized OT barycenter for $c_k(x, y) = \frac{1}{2}\|x - y\|^2$ is itself Gaussian and can be computed using the well-celebrated fixed point iterations of (Álvarez-Esteban et al., 2016, Eq. (19)). This gives us an opportunity to compare our results

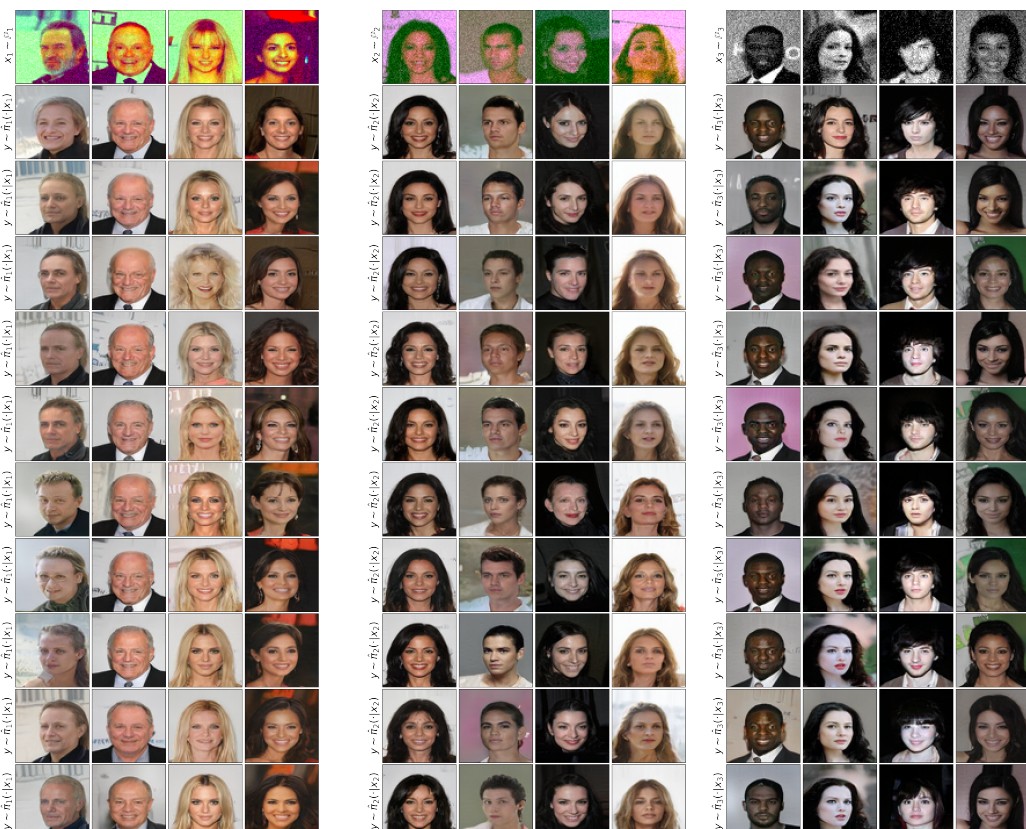

(a) Maps from $\mathbb{P}_1$ to the barycenter. (b) Maps from $\mathbb{P}_2$ to the barycenter. (c) Maps from $\mathbb{P}_3$ to the barycenter.

Figure 6: *Experiment on the Ave, celeba! barycenter dataset.* The plots show additional examples of samples transported with **our** solver to the barycenter.

with the ground-truth unregularized barycenter in the Gaussian case. As the **baseline**, we include the results of (Korotin et al., 2022a, WIN) solver which learns the unregularized barycenter ($\epsilon = 0$).

We consider 3 Gaussian distributions $\mathbb{P}_1, \mathbb{P}_2, \mathbb{P}_3$ in dimensions $D = 2, 4, 8, 16, 64$ and compute the approximate EOT barycenters $\pi_k^{\hat{f}_k}$ for $\epsilon = 0.01, 1$ w.r.t. weights $(\lambda_1, \lambda_2, \lambda_3) = (\frac{1}{4}, \frac{1}{4}, \frac{1}{2})$ with our solver. To initialize these distributions, we follow the strategy of (Korotin et al., 2022a, Appendix C.2). The ground truth unregularized barycenter $\mathbb{Q}^*$ is estimated via the above-mentioned iterative procedure. We use the code from WIN repository available via the link mentioned in Appendix C.2. To assess the WIN solver, we use the unexplained variance percentage metrics defined as $\mathcal{L}_2\text{-UVP}(\hat{T}) = 100 \cdot [\|\|\hat{T} - T^*\|\|_{\mathbb{P}}^2$ where $T^*$ denotes the optimal transport map $T^*$, see (Korotin et al., 2021a, §5.1). Since our solver computes EOT plans but not maps, we evaluate the barycentric projections of the learned plans, i.e., $\widehat{\overline{T}}_k(x) = \int_{\mathcal{Y}} y \pi_k^{\hat{f}_k}(y|x)$, and calculate $\mathcal{L}_2\text{-UVP}(\widehat{\overline{T}}_k, T_k^*)$. We evaluate this metric using $10^4$ samples. To estimate the barycentric projection in our solver, we use $10^3$ samples $y \sim \pi_k^{\hat{f}_k}(y|x_k)$ for each $x_k$. To keep the table with the results simple, in each case we report the average of this metric for $k = 1, 2, 3$ w.r.t. the weights $\lambda_k$.

| Dim / Method | Ours ($\epsilon = 1$) | Ours ($\epsilon = 0.01$) | (Korotin et al., 2022a, WIN) |
|---|---|---|---|
| 2 | 1.12 | **0.02** | 0.03 |
| 4 | 1.6 | **0.05** | 0.08 |
| 8 | 1.85 | **0.06** | 0.13 |
| 16 | 1.32 | **0.09** | 0.25 |
| 64 | 1.83 | 0.84 | **0.75** |

Table 4: $\mathcal{L}_2$-UVP for our method with $\epsilon = 0.01, 1$ and WIN, $D = 2, 4, 8, 16, 64$.

**Results.** We see that for small $\epsilon = 0.01$ and dimension up to $D = 16$, our algorithm gives the results even better than WIN solver designed specifically for the unregularized case ($\epsilon = 0$). As was expected, larger $\epsilon = 1$ leads to the increased bias in the solutions of our algorithm and $\mathcal{L}_2$-UVP metric increases.

## D ALTERNATIVE EBM TRAINING PROCEDURE

In this section, we describe an alternative **simulation-free** training procedure for learning EOT barycenter distribution via our proposed methodology. The key challenge behind our approach is to estimate the gradient of the dual objective (3). To overcome the difficulty, in the main part of our manuscript, we utilize MCMC sampling from conditional distributions $\mu_{x_k}^{f_{\theta,k}}$ and estimate the loss with Monte-Carlo. Here we discuss a potential alternative approach based on **importance sampling** (IS) (Tokdar & Kass, 2010). That is, we evaluate the internal integral over $\mathcal{Y}$ in (3):

$$\mathcal{I}(x_k) \stackrel{\text{def}}{=} \int_{\mathcal{Y}} \left[ \frac{\partial}{\partial \theta} f_{\theta,k}(y) \right] \mathrm{d}\mu_{x_k}^{f_{\theta,k}}(y) \tag{36}$$

with help of an auxiliary proposal (continuous) distribution accessible by samples with the known density $q(y)$. Let $Y^q = \{y_1^q, \ldots, y_P^q\}$ be a sample from $q(y)$. Define the weights:

$$\omega_k(x_k, y_p^q) \stackrel{\text{def}}{=} \exp\left( \frac{f_{\theta,k}(y_p^q) - c(x_k, y_p^q)}{\varepsilon} \right) q(y_p^q).$$

Then (36) permits the following stochastic estimate:

$$\mathcal{I}(x_k) \approx \frac{\sum_{p=1}^{P} \left[ \frac{\partial}{\partial \theta} f_{\theta,k}(y) \right] \omega_k(x_k, y_p^q)}{\sum_{p=1}^{P} \omega_k(x_k, y_p^q)}. \tag{37}$$

**Experimental illustration.** To demonstrate the applicability of IS-based training procedure to our barycenter setting, we conduct the experiment following our **2D Twister** setup, see §5.1. We employ zero-mean $16I$-covariance Gaussian distribution as $q$ and pick the batch size $P = 1024$. Our results are shown in Figure 7. As we can see, the alternative training procedure yields similar results to Figure1 but converges faster ($\approx 1$ min. VS $\approx 18$ min. of the original MCMC-based training).

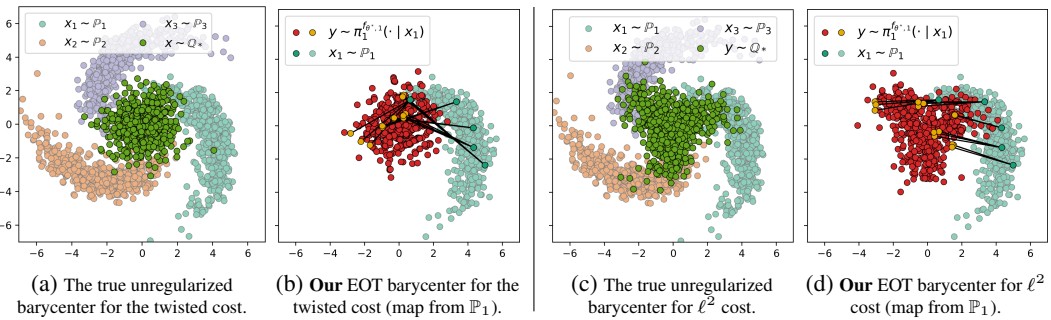

(a) The true unregularized barycenter for the twisted cost.

(b) **Our** EOT barycenter for the twisted cost (map from $\mathbb{P}_1$).

(c) The true unregularized barycenter for $\ell^2$ cost.

(d) **Our** EOT barycenter for $\ell^2$ cost (map from $\mathbb{P}_1$).

Figure 7: *2D twister example. **Trained with importance sampling**:* The true barycenter of 3 comets vs. the one computed by our solver with $\epsilon = 10^{-2}$. Two costs $c_k$ are considered: the twisted cost (7a, 7b) and $\ell^2$ (7c, 7d). We employ the *simulation-free* importance sampling procedure for training.

**Concluding remarks.** We note that IS-based methods requires accurate selection of the proposal distribution $q$ to reduce the variance of the estimator (Tokdar & Kass, 2010). It may be challenging in real-world scenarios. We leave the detailed study of more advanced IS approaches in the context of energy-based models and their applicability to our EOT barycentric setup to follow-up research.

