# OpenReview forum: "Energy-Guided Continuous Entropic Barycenter Estimation for General Costs"
_ICLR.cc/2024/Conference — Submitted to ICLR 2024_

### Official Review · Reviewer_97mW · 2023-10-31

**Soundness:** 3 good
**Presentation:** 3 good
**Contribution:** 2 fair
**Rating:** 6
**Confidence:** 4

**Summary:**

This paper considers the problem of computing entropic optimal transport barycenters from continuous distributions. It is assumed that the authors only observe these continuous distributions from samples. The approach involves studying the dual of the entropic OT problem and using neural networks to approximate dual potentials. The barycenter can then be recovered from these potentials using the primal-dual relationship. With this new formulation, the authors can compute barycenters for general costs, which allows for the computation of barycenters under learned geometry (via StyleGAN). The method is also scalable to larger datasets than existing methods.

**Strengths:**

- The authors use a conditional formulation of the entropic barycenter problem to give a new dual formulation.  They also give a stability bound, which gives an error bound on the quality of the dual potentials in terms of the primal solution.
- The authors also give learning bounds for barycenters in terms of the loss and estimation error. They also give a universal approximation result that states that there exist neural networks that can achieve this bound. The proof of duality and quality of dual potentials is essential.
- Because the method works with general costs, it can work with manifold constraints, which gives some nice results in Sections 5.2 and 5.3. This is perhaps the most significant contribution of the work.

**Weaknesses:**

My concerns mostly center around 1) the impact this work can have and 2) how many new ideas it contributes to the literature. While the extension to general costs is interesting, it seems somewhat straightforward, and I don't see how the methods developed themselves are necessarily tailored to the general cost nature of the problem.

- The theorems all seem to have standard proofs that are straightforward extensions of existing results. I am not sure that the theory proposed gives any new techniques or insights. Furthermore, more important proofs are missing, such as how efficiently one can sample from the barycenter. Furthermore, the finite sample learning guarantees are nice to see but have no practical impact since there are no cases where they could estimate these errors in any practical example.
- The method learns OT plans but not maps. This is true of the general entropically regularized setting. However, this adds complexity to the usefulness of the method because one doesn’t have access direction to the barycenter distribution itself and rather only has conditional access through the reference measures.
- In Figure 3, the method proposed doesn’t recover the image content due to its conditional nature, despite the fact that their FID is lower. Is there any way of getting samples closer to the truth? Also, I guess the FID is lower than other existing methods because of the addition of the StyleGAN.
- The samples, even in the manifold-constrained case in Figure 5.2, are comparable to SCWB, and therefore it is hard to see why the proposed method is better. This experiment is qualitative but not quantitative.
- The authors acknowledge this, but it is difficult to compare with the true entropic optimal transport barycenter quantitatively. However, it would have been nice to see it in cases where it is known (for example, with Gaussians).

**Questions:**

- In training, how do you ensure that the ULA is running enough for convergence?
- Can the authors comment on the complexity of the method in relation to the others? Is it really that the proposed method is more scalable, or just that they test on larger datasets?

---

> ### Author Response · Authors · 2023-11-21
> **Answer to reviewer 97mW. Part 1**
>
> Dear reviewer, thank you for spending time reviewing our paper. Please take a look at our general answer to all the reviewers where we address common questions and show additional experimental results. Below we answer to your questions and comments.
>
> **(0) The authors acknowledge this, but it is difficult to compare with the true entropic optimal transport barycenter quantitatively. However, it would have been nice to see it in cases where it is known (for example, with Gaussians).**
>
> See the general answer to all reviewers for the discussion of this question and newly added comparison with ground-truth *unregularized* barycenter.
>
> **(1) The theorems all seem to have standard proofs that are straightforward extensions of existing results. I am not sure that the theory proposed gives any new techniques or insights.**
>
> Thank you for this opinion. However, from our humbly point of view, this is not that right.
>
> (a) **"standard proofs [...] straightforward extensions[...]"**.
>
> While our Theorems 2, 3, 4 are indeed largely based on the similar results from [4], we argue that our Theorem 1 (the key result for our proposed methodology) is original, and its proof is non-trivial. We have even been forced to prove some auxiliary properties of weak $C$-transform. The latter are interesting on its own.
>
> (b) **"[...] not sure that the theory gives new techniques or insights."**
>
> Our developed theory allows us to build Entropic OT barycenter solver for general cost function based on extensively developed EBMs techniques. We think this is a worthy result of our theoretical deductions. The closest approach [4] develops Energy-based training procedures only for Entropic OT problem itself. It is related but different problem. We do not hide our connection to [4], but we take a considerable step forward from both a theoretical and a practical perspectives.
>
> **(2)   Furthermore, more important proofs are missing, such as how efficiently one can sample from the barycenter. Furthermore, the finite sample learning guarantees are nice to see but have no practical impact since there are no cases where they could estimate these errors in any practical example.**
>
> (a) **efficient sampling from the barycenter**
>
> Our proposed practical algorithm is  based on Energy-based models. It is an autonomous decades-long research direction which comes with its own advantages, techniques and limitations. The question of efficient sampling from barycenters resorts to the similar question for general-purpose EBMs. We point the reviewer's attention to [8], [9], [10] for some attempts to improve the sampling characteristics of EBMs. Note that typically these methods follows from practical observations, not theorems.
>
> In our paper, we also do not make an attempt to equip EBMs with theoretically-grounded efficient sampling method. In fact, we treat the EBM training procedure as the particular solver for our proposed methodology. We leave the question regarding efficient sampling techniques to a separate research.
>
> (b) **"finite sample learning guarantees [...] no practical impact"**
>
> We agree that the particular statement of our theorem is not directly related to practical procedure, just like many Statistical Machine Learning researches in Generative Modelling. At the same time, our result justifies that our methodology actually permits the particular Statistical and Approximation guarantees. Note that deriving the similar guarantees for other scalable continuous barycenter solvers is complicated. This is because they utilize nontrivial adversarial [2]/iterative [1]/diffusion-based [7] objectives, while our objective (up to MCMC procedure) is the direct minimization of sum of KL divergences (see Theorem 2).
>
>
> **(3)  The method learns OT plans but not maps. This is true of the general entropically regularized setting. However, this adds complexity to the usefulness of the method because one doesn’t have access direction to the barycenter distribution itself and rather only has conditional access through the reference measures.**
>
> While we partially agree with the reviewer, to be honest, we do not think this is a problem. Indeed, one can learn generative model (e.g., a GAN or a diffusion) on top of samples transported with our solver from marginal distributions to the learned barycenter. This will provide the direct access to samples of barycenter. Moreover, in the **manifold-constrained** case (Sections 5.2, 5.3), such generative model has to be constructed in the latent space (of a StyleGAN) which is presumably even easier.

---

> > ### Author Response · Authors · 2023-11-21
> > **Answer to reviewer 97mW. Part 2**
> >
> > **(4) In Figure 3, the method proposed doesn’t recover the image content due to its conditional nature, despite the fact that their FID is lower. Is there any way of getting samples closer to the truth? Also, I guess the FID is lower than other existing methods because of the addition of the StyleGAN.**
> >
> >
> > When $\epsilon\rightarrow 0$, the EOT barycenter becomes closer to the unregularized one. Hence, one can **get samples closer to the ground truth** by picking even smaller $\epsilon$ that we used. However, two *computational* challenges appear. First, the conditional Langevin sampling in our algorithm becomes less stable as $\epsilon \to 0$ because the magnitude of $\nabla \frac{f_{k}(y) - c_{k}(x_k,y)}{\epsilon}$ gets bigger. Second, the energy landscape $\pi^{f_k}(y|x_k)\propto \exp(\frac{f_{k}(y)-c_{k}(x_k,y)}{\epsilon})$ becomes more peaked and hence challenging. The situation is very analogous to that in the well-celebrated Sinkhorn algorithm [3] for discrete OT.
> >
> > **Regarding the FID score**, StyleGAN indeed helps to improve FID. That is, our methods allows to combine the barycenter estimation with well-performing developments from the field of the generative modeling. We think this is a notable advantage since most other solvers can not do this as this requires using general cost functions.
> >
> > **(5) The samples, even in the manifold-constrained case in Figure 5.2, are comparable to SCWB, and therefore it is hard to see why the proposed method is better. This experiment is qualitative but not quantitative.**
> >
> > **Remark.** First of all, we would like to apologize for the misprint because we accidentally swapped the labels $\textbf{SCWB}$ and $\textbf{WIN}$ in the corresponding Figures 2a and 2b. It seems like it is the WIN method that you talk about. Second, there is an another misprint: we accidentally added the wrong image for **Our (manifold-constrained)** maps $1\rightarrow 0$. Fixed now.
> >
> > **Answer.** Since the previous barycenter papers [1, 2] consider this qualitative experiment, we also repeat it. The goal here is to demonstrate the visual quality and the preservation of the content of the input sample, i.e., qualitatively check the optimality of the map to the barycenter. We agree with the reviewer that the experiment is purely  qualitative, however, it is caused by absence of the known ground-truth barycenter.
> >
> > In order to demonstrate the quantitative advantages of our method, we conduct the experiment on "Ave, Celeba!" dataset (see Section 5.3) with the known ground-truth unregularized barycenter (see Section 5 of [1]). It is much more complicated as it considers $64\times 64\times 3$ images instead of grayscale $32\times 32.$
> >
> > **(6) In training, how do you ensure that the ULA is running enough for convergence?**
> >
> > We follow the basic and well-established practices for running the sampling procedure. In general, there are no guarantees of convergence, however, our newly conducted experiments dealing with barycenters of Gaussian distributions (**Appendix C.4**) demonstrate the convergence of our solver to true barycenter. Indirectly, this testifies the convergence of ULA. Additionally, we draw the reviewer's attention to the fact that EBMs which utilize ULA-like training procedures demonstrate high-quality results in Generative Modelling.
> >
> > **(7) Can the authors comment on the complexity of the method in relation to the others?**
> >
> > As we wrote in our paper, the MCMC procedure might be time-consuming.
> > At the same time, in comparison to most other solvers, our method scales better and works for general costs, see the next answer.

---

> > > ### Author Response · Authors · 2023-11-21
> > > **Answer to reviewer 97mW. Part 3**
> > >
> > > **(8) Is it really that the proposed method is more scalable, or just that they test on larger datasets?**
> > >
> > > Our developed solver allows employing pre-trained generative models (e.g., the StyleGAN) to constrain the search space for the barycenter. This helps to improve the scalability compared to the predecessors, as predicting the latent code is much simpler that generating the image. As we already noted above, most other solvers do not support this feature because they mostly work with the quadratic cost. Please see the discussion at the end of Section 3.
> > >
> > > **Concluding remarks.** Please reply to our post and inform us if the clarifications provided adequately address your concerns regarding our work. We are more than willing to discuss any remaining points during the discussion phase. If the responses offered meet your satisfaction, we kindly request you to consider raising your score.
> > >
> > > **Additional references.**
> > >
> > > [1] Alexander Korotin, Vahe Egizarian, Lingxiao Li and Evgeny Burnaev. Wasserstein Iterative Networks for Barycenter Estimation. In NeurIPS, 2022.
> > >
> > > [2] Jiaojiao Fan, Amirhossein Taghvaei, Yongxin Chen.Scalable Computations of Wasserstein Barycenter via Input Convex Neural Networks. In ICML, 2021.
> > >
> > > [3] Marco Cuturi .Sinkhorn Distances: Lightspeed Computation of Optimal Transport. In NeurIPS, 2013.
> > >
> > > [4] Mokrov et. al., Energy-guided Entropic Neural Optimal Transport.
> > >
> > > [5] Yang Zhao, Jianwen Xie, Ping Li.Learning. Energy-Based Generative Models via coarse-to-fine expanding and sampling. In ICLR, 2021.
> > >
> > > [6] Yang Song, Diederik P. Kingma. How to Train Your Energy-Based Models. arXiv:2101.03288
> > >
> > > [7] Maxence Noble, Valentin De Bortoli, Arnaud Doucet and Alain Durmus. Tree-Based Diffusion Schr$\backslash$" odinger Bridge with Applications to Wasserstein Barycenters. In NeurIPS, 2023.
> > >
> > > [8] Gao et. al., Learning Energy-Based Models by Diffusion Recovery Likelihood. arXiv:2012.08125
> > >
> > > [9] Zhao et. al., Learning Energy-Based Generative Models via Coarse-to-Fine Expanding and Sampling. In ICLR, 2020.
> > >
> > > [10] Du et. al., Improved Contrastive Divergence Training of Energy-Based Models. arXiv:2012.0131

---

> > > > ### Comment · Reviewer_97mW · 2023-11-22
> > > >
> > > > I appreciate the authors detailed response to my review, and while there may be some disagreement on the usefulness/novelty of the theoretical results, I’m inclined to raise my score.

---

### Official Review · Reviewer_BAkA · 2023-10-31

**Soundness:** 3 good
**Presentation:** 3 good
**Contribution:** 2 fair
**Rating:** 5
**Confidence:** 3

**Summary:**

In this paper, the authors propose a new algorithm for approximating the entropy-regularized optimal transport barycenter of continuous probability distributions with general cost functions. In particular, they derive the weak dual formulation of the original problem, and then leverage energy-based models for an optimization procedure. Additionally, they also establish the generalization bounds and universal approximation guarantees for their estimation of optimal transport plan. Finally, they justify the efficacy of their proposed method by conducting some experiments on generative models.

**Strengths:**

1.Although the idea of leveraging Energy-based models to solve optimal transport problems is not novel as it was already introduced in [1, 2], its applications on the barycenter problem is new.

2. The results presented in the paper have been both theoretically and empirically demonstrated, which help strengthen the paper.

3. The paper is well-written with no grammatical errors (as far as I can tell).

**References**

[1] Petr Mokrov, Alexander Korotin, Evgeny Burnaev. Energy-guided Entropic Neural Optimal Transport.

[2] Khai Nguyen and Nhat Ho. Energy-Based Sliced Wasserstein Distance. In NeurIPS, 2023.

**Weaknesses:**

1. The authors only motivated the optimal transport barycenter problem in Section 1, but they did not explain why its entropy-regularization variant is worth discovering.

2. The MCMC algorithm leveraged in Algorithm 1 is not efficient.

3. There are some incorrect claims in the paper (see Question 2).

4. The paper organization is not good:
- The authors should move the notation paragraph in Section 2 to the end of Section 1, which makes more sense.
- In Appendix A, the authors should separate the proofs of theoretical result into different subsections, which makes it easier for readers to navigate.

5. The paper violates the 9-page rule of ICLR 2024.

**Questions:**

1. Does the result in equation (6) hold true for any continuous function $f\in\mathcal{C}(\mathcal{Y})$ or only for the optimal function $f$ in equation (3)?

2. At the beginning of Section 2.3, the authors claimed that there was no direct analytical solution for the entropic barycenter of Gaussian distributions. However, Le et al. [2] already arrived at the closed-form expression for that problem with the inner-product cost function (see Theorem 5.2).

3. If the input distributions $\mathbb{P}_1,\ldots,\mathbb{P}_K$ are not probability distributions and do not share the same mass, would we be able to extend the enery-guided approach to approximate the barycenter of those distributions?

4. If we replace the MCMC algorithm with other sampling methods, namely importance sampling, would the running time be improved? If not, what sampling methods are possible to achieve an improved running time. The authors should discuss more about this in Section 6.

5. The authors should cite more relevant papers:
- In Section 2.2, the entropic barycenter problem was also previously studied in [1, 2];

6. Typos:
- Below equation (8), I guess the term 'which differs from (8)' should be 'which differs from (7)'. Please correct me if I was wrong.
- Below equation (8), the author should clarify the abbreviation l.s.c although I can guess that it stands for lower semi-continuous.

**References**

[1] Khang Le, Huy Nguyen, Quang Nguyen, Tung Pham, Hung Bui, Nhat Ho. On Robust Optimal Transport: Computational Complexity and Barycenter Computation. In NeurIPS, 2021.

[2] Khang Le, Dung Le, Huy Nguyen, Dat Do, Tung Pham, Nhat Ho. Entropic Gromov-Wasserstein between Gaussian Distributions. In ICML, 2022.

---

> ### Author Response · Authors · 2023-11-21
> **Answer to Reviewer BAkA (1/2)**
>
> Dear reviewer, thank you for spending time reviewing our paper. Please take a look at our general answer to all the reviewers where we address common questions and show additional experimental results. Below we answer to your questions and comments.
>
> **(1) The authors only motivated the optimal transport barycenter problem in Section 1, but they did not explain why its entropy-regularization variant is worth discovering.**
>
> The logic behind the introduction of entropy is analogous to its introduction in case of discrete OT. Considering entropy-regularized problem in *discrete* case allows to
>
> **(a)** provide improved theoretical properties such as the strict convexity of the OT/barycenter problem, uniqueness of the solution, derive the tractable OT dual form [1] which, in turn, allows to
>
> **(b)** establish convenient algorithms such as the Sinkhorn-like OT/barycenter methods with guarantees of performance.
>
> The situation in the **continuous** case is the same. Point **(a)** still holds, and, **as we show in our paper**, the tractable OT dual form (Eq. 3) for EOT **(b)** gives us the opportunity to apply the well-known techniques from Energy-based modeling (EBM) to solve the barycenter problem under consideration.
>
> Furthermore, our established solver allows to use general cost functions $c_k$ and, in particular, permits employing pre-trained generative models (e.g., StyleGAN) to restrict the sought-for barycenter to the image manifold. This aspect contributes to scalability of the proposed algorithm and expands practical applications of the barycenter problem.
>
> **(2) The MCMC algorithm leveraged in Algorithm 1 is not efficient. [...]  If we replace the MCMC algorithm with other sampling methods, namely importance sampling, would the running time be improved? If not, what sampling methods are possible to achieve an improved running time. The authors should discuss more about this in Section 6.**
>
> We would like to note that we develop a generic **methodology** for solving the EOT barycenter problem and demonstrate that it does work even with the simplest sampling procedures. In principle, importance sampling might be used as such procedure (instead of ULA), however, it scales poorly with dimensionality (see Section 3.3 of [2]) and demands  the usage of advanced improvements. Thus,  the result of generating potentially might be improved with utilization either more efficient MCMC algorithms [5] or advanced  techniques for importance sampling procedures [2, 6], however, these considerations are beyond the scope of our paper.
>
> Following your suggestion, we mentioned corresponding papers in Section 6.
>
> **(3) Does the result in equation (6) hold true for any continuous function $f \in \mathcal{C}(\mathcal{Y})$  or only for the optimal function  $f$
>  in equation (3)?**
>
> The results holds for every $f \in \mathcal{C}(\mathcal Y)$. If $f$ is optimal ($f=f^{*}$), then Equation 6 turns to  the equality $0 = 0$.
>
> **(4) If the input distributions  $\mathbb{P}_1,..,\mathbb{P}_k$ are not probability distributions and do not share the same mass, would we be able to extend the energy-guided approach to approximate the barycenter of those distributions?**
>
> Adapting our approach to distributions of unequal mass (*unbalanced* setup) is beyond the scope of our current paper. It would require significant changes in methodology and theoretical foundations. Exploring this direction is an interesting avenue for future research.
>
>
> **(6) paper organization is not good: [...] move the notation paragraph in Section 2 to the end of Section 1. [...] In Appendix A, the authors should separate the proofs of theoretical result into different subsections.**
>
> Up to the reviewer's request, we moved the notation and separated the proofs in the revised version of our paper.
>
> **(7) The paper violates the 9-page rule of ICLR 2024.**
>
> Please note that according to ICLR 2024 rules, it is allowed to position the reproducibility statement on an additional 10th page, see the reproducibility section in https://iclr.cc/Conferences/2024/AuthorGuide
>
> **(8) The authors should cite more relevant papers. In Section 2.2, the entropic barycenter problem was also previously studied in papers [3, 4].**
>
> We thank the reviewer for pointing to the relevant papers [3, 4]. We added the citations in Section 2.2, Appendix B.1 and C.2.
>
> **(9) Typos:
> Below equation (8), the term 'which differs from (8)' should be 'which differs from (7)', [...] clarify the abbreviation l.s.c although I can guess that it stands for lower semi-continuous.**
>
> We thank the reviewer for indicating the typos, we fixed them in the revised version of our paper.
>
> **Concluding remarks.** Please reply to our post and inform us if the clarifications provided adequately address your concerns regarding our work. We are more than willing to discuss any remaining points during the discussion phase. If the responses offered meet your satisfaction, we kindly request you to consider raising your score.

---

> > ### Author Response · Authors · 2023-11-21
> > **Answer to Reviewer BAkA (2/2)**
> >
> > **References.**
> >
> > [1] Marco Cuturi .Sinkhorn Distances: Lightspeed Computation of Optimal Transport. In NeurIPS, 2013.
> >
> > [2] Dieterich Lawson, George Tucker, Bo Dai, Rajesh Ranganath. Energy-Inspired Models: Learning with Sampler-Induced Distributions. In NeurIPS, 2019.
> >
> > [3] Khang Le, Huy Nguyen, Quang Nguyen, Tung Pham, Hung Bui, Nhat Ho. On Robust Optimal Transport: Computational Complexity and Barycenter Computation. In NeurIPS, 2021.
> >
> > [4] Khang Le, Dung Le, Huy Nguyen, Dat Do, Tung Pham, Nhat Ho. Entropic Gromov-Wasserstein between Gaussian Distributions. In ICML, 2022.
> >
> > [5] Yilun Du, Conor Durkan, Robin Strudel, Joshua B. Tenenbaum, Sander Dieleman, Rob Fergus,
> > Jascha Sohl-Dickstein, Arnaud Doucet, Will Grathwohl. Reduce, Reuse, Recycle: Compositional Generation with Energy-Based Diffusion Models and MCMC. In ICML,2023
> >
> > [6] Meng Liu, Haoran Liu, Shuiwang Ji. Gradient-Guided Importance Sampling for Learning Discrete Energy-Based Models.

---

> > > ### Comment · Reviewer_BAkA · 2023-11-22
> > >
> > > Dear Authors,
> > >
> > > Thanks for your response, which addresses some of my concerns. However, there are still many other quesitons which have not been addressed yet. In particular,
> > >
> > > 1. The discussion about other possibly efficient sampling methods than MCMC is currently quite short, and there are neither intuitions nor insights of how to improve the sampling procedures. I suggest that the authors should study other sampling methods for EOT barycenter more thoroughly on both theoretical and empirical sides, which strengthens the paper substantially.
> > >
> > > 2. The authors does not show effort to answer my question about the unbalanced settings of the EOT barycenter problem. I would like to emphasize that those settings are very important in making the optimal solution robust to outliers which often appears in real-world datasets.
> > >
> > > For those reasons, I think my final score of 5 is reasonable, and I decide to keep it.
> > >
> > > Best,
> > >
> > > Reviewer BAkA

---

> > > > ### Author Response · Authors · 2023-11-23
> > > > **Additional Response to Reviewer BAkA**
> > > >
> > > > Dear reviewer, thanks for responding to our answers and further engaging the discussion.
> > > >
> > > > **(1) The discussion about other possibly efficient sampling methods than MCMC (...), there are neither intuitions nor insights of how to improve the sampling procedures. I suggest that the authors should study other sampling methods for EOT barycenter (...) which strengthens the paper substantially.**
> > > >
> > > > On the one hand, we humbly believe that the detailed analysis of efficient MCMC methods are beyond the scope of our method. At the same time, following your suggestion, we add some *insights* as well as *supporting experiment* which illustrate the alternative simulation-free **Importance sampling** based approach for optimizing our proposed objective. See newly added Appendix D (text color is *orange*). We also updated our code in the supplementary materials. See newly added `notebooks//toy_2d_twister_importance.ipynb` notebook in the supplementary archive.
> > > >
> > > > **(2) (...) unbalanced settings of the EOT barycenter problem. I would like to emphasize that those settings are very important in making the optimal solution robust to outliers which often appears in real-world datasets.**
> > > >
> > > > We agree with the reviewer that the unbalanced setting of the barycenter problem is important in several real-world scenarios. However, as far as we know, existing approaches to *unbalanced* OT/EOT barycenter problem [1,3,4,7] tackle discrete case while our EOT barycenter solver is designed for the *continuous setup* which opens up a *new scope of potential applications in real-world tasks*. Additionally, unlike current continuous OT barycenters [2, 5, 6] our solver is unique in that it allows to use *general cost functions* which significantly boosts its practical usability.
> > > >
> > > > In essense, (1) we agree that unbalanced setting of our approach is an interesting avenue for future research; (2) we argue that **our solver is of interest by itself**, see Appendix B.2 for a description of potential applications.
> > > >
> > > > **References.**
> > > >
> > > > [1] Chizat, L., Peyré, G., Schmitzer, B., & Vialard, F. X. (2018). Scaling algorithms for unbalanced optimal transport problems. Mathematics of Computation, 87(314), 2563-2609.
> > > >
> > > > [2] Fan, J., Taghvaei, A., & Chen, Y. (2021, July). Scalable Computations of Wasserstein Barycenter via Input Convex Neural Networks. In International Conference on Machine Learning (pp. 1571-1581). PMLR.
> > > >
> > > > [3] Heinemann, F., Klatt, M., & Munk, A. (2023). Kantorovich–Rubinstein Distance and Barycenter for Finitely Supported Measures: Foundations and Algorithms. Applied Mathematics \& Optimization, 87(1), 4.
> > > >
> > > > [4] Le, K., Nguyen, H., Nguyen, Q. M., Pham, T., Bui, H., & Ho, N. (2021). On robust optimal transport: Computational complexity and barycenter computation. Advances in Neural Information Processing Systems, 34, 21947-21959.
> > > >
> > > > [5] Li, L., Genevay, A., Yurochkin, M., & Solomon, J. M. (2020). Continuous regularized wasserstein barycenters. Advances in Neural Information Processing Systems, 33, 17755-17765.
> > > >
> > > > [6] Korotin, A., Egiazarian, V., Li, L., & Burnaev, E. (2022, May). Wasserstein Iterative Networks for Barycenter Estimation. In Advances in Neural Information Processing Systems.
> > > >
> > > > [7] Schmitzer, B. (2019). Stabilized sparse scaling algorithms for entropy regularized transport problems. SIAM Journal on Scientific Computing, 41(3), A1443-A1481.

---

### Official Review · Reviewer_7U49 · 2023-10-31

**Soundness:** 4 excellent
**Presentation:** 4 excellent
**Contribution:** 3 good
**Rating:** 8
**Confidence:** 3

**Summary:**

The paper proposes a novel algorithm for approximating the continuous Entropic Optimal Transport (EOT) barycenter for arbitrary Optimal Transport (OT) cost functions. The approach is based on the dual reformulation of the EOT problem based on weak OT, which has recently gained the attention of the ML community. Various advantages of this method were proposed. Some real-world applications in ML are used to demonstrate the effectiveness of the proposed method.

**Strengths:**

This is a mathematically heavy paper, and the authors presented it quite well. The idea is to first formulate the EOT barycentric problem into a dual formulation (Theorem 1). With the dual formulation, a practical optimization algorithm is proposed via approximating the potentials used in the dual formulation by neural network functions. Given the concrete neural network approximator, the gradient of the EOT barycenter objective can be explicitly written. Based on this, the authors propose an algorithm for the EOT optimization.

**Weaknesses:**

The results demonstrate generalization error estimates and universal approximation via neural network results for their algorithm. However, this part is a bit too general. I wonder if there are more specific results for special kind of neural network towards their approximation quality.

**Questions:**

see "Weaknesses" section.

---

> ### Author Response · Authors · 2023-11-21
> **Answer to reviewer 7U49**
>
> Dear reviewer, thanks for your positive feedback.
>
> **(1)The results demonstrate generalization error estimates and universal approximation via neural network results for their algorithm. However, this part is a bit too general. I wonder if there are more specific results for special kind of neural network towards their approximation quality.**
>
> Improving Theorem 4 with explicit numerical bounds is indeed a good question coming from Statistical Learning Theory. However, it seems to be rather non-trivial and deserves a separate research, probably, a separate paper. The key challenge here is the analysis of weak $C_{\text{EOT}}$-transformed functions $f^{C}$. This is much more difficult than, say, analysis of just functions (Neural Networks) $f$. Therefore, we expect that the derivation of the desired bounds will require substantial theoretical and numerical efforts: definitions, assumptions, theorems, proofs and supporting experiments. We leave all this stuff to a follow-up research.
>
>
> **Concluding remarks.** Please reply to our post and inform us if the clarifications provided adequately address your concerns regarding our work. We are more than willing to discuss any remaining points during the discussion phase.

---

> > ### Comment · Reviewer_7U49 · 2023-11-23
> > **keep my score**
> >
> > Thanks for the response. I would like to keep my score as it was.

---

### Official Review · Reviewer_16og · 2023-11-07

**Soundness:** 3 good
**Presentation:** 3 good
**Contribution:** 3 good
**Rating:** 6
**Confidence:** 3

**Summary:**

In this work, the authors propose a new algorithm for computing the entropic-regularized Wasserstein barycenter for arbitrary cost functions with theoretical guarantees. The proposed algorithm is based on the dual formulation of the entropic OT problem. An integral involved in the proposed algorithm is approximated using an MCMC procedure (unadjusted Monte Carlo). Extensive experiments are performed to illustrate the usefulness of the proposed method.

**Strengths:**

A new algorithm for computing the entropic-regularized Wasserstein barycenter for arbitrary cost functions is proposed, which seems interesting, with a lot of empirical evidence.

**Weaknesses:**

Most theoretical results here seem trivial or a bit irrelevant to me. Theorem 2 appears to follow from the definitions, while Theorems 4 and 5 do not seem to be very relevant to the proposed algorithm. I would expect more discussion on the convergence behavior of the proposed algorithm say convergence rates under simple settings but that is lacking in the paper.

---
After rebuttal: the authors have explained my concerns above; see below.

**Questions:**

Could you answer how relevant Theorems 4 and 5 are towards the understanding of the proposed algorithm? In addition, from my knowledge of the literature, it does not appear to be common to use neural networks for parameterization of the potentials (correct me if I am wrong). How much computational overhead this will incur on the computation, especially if $K$ is large?
The mention of energy-based models (EBMs) does not seem to be very relevant to me either. Ultimately an MCMC procedure such as ULA is used to generate samples in order to approximate an integral. EBMs are related but not that relevant. Also, the authors appear to have used ULA with stochastic/batch gradients, which is better known as stochastic gradient Langevin dynamics (SGLD).

Typo: page 17: swap $\min$ and “$\sup$” in (17) …

---

> ### Author Response · Authors · 2023-11-21
> **Answer to reviewer 16og (1/2)**
>
> Dear reviewer, thank you for spending time reviewing our paper.
>
> **(1) Trivial theoretical results. "Theorem 2 appears to follow from the definitions".**
>
> At first, we highlight that the key theoretical result of our work is Theorem 1. From our point of view, it is not a trivial result. Secondly, we agree that proving Theorem 2 is not a difficult task. At the same time, we argue that Theorem 2 is *important and worth presenting*. It states that the gap between optimal value of (dual) training objective $\mathcal{L}$ and an actual value *directly* bounds the quality of the recovered barycentric plans $\pi^{f_k}$. It leads to the straightforward conclusion that the closer is our optimized objective $\mathcal{L}$ to the optimal one, the more accurate are the recovered barycentric plans $\pi^{f_k}$. Combined with Theorem 1, this result forms the **foundation** of our approach.
>
>
> **(2) Convergence speed/rates discussions. "[...] more discussion on the convergence behavior of the proposed algorithm say convergence rates under simple settings but that is lacking in the paper."**
>
> We thank the reviewer for this interesting question. Let us formalize the problem a bit. In Section $\S$ 4.3 we introduced the set of **perfect** potentials $(\widehat{f_1}, \dots \widehat{f_K})$ which solve the entropic barycenter problem given limited number of empirical samples and limited class of available potentials. If we correctly understand, your question is about how fast the **optimized** potentials $(\widehat{f_{1,opt}} , \dots \widehat{f_{K,opt}})$ converge to the perfect ones. Let us call the discrepancy between optimized and perfect potentials the *optimization* error. There are multiple sources of *optimization* error and there are multiple factors which affect its dynamics: inexact sampling due to MCMC and stochastic gradient ascent (Eq. 12). The analysis of these quantities is a completely different domain in Machine Learning and is out of scope of our work. As with most generative modeling research, we do not attempt to analyze optimization errors.
>
> **(3) "Could you answer how relevant Theorems 4 and 5 are towards the understanding of the proposed algorithm?"**
>
> Continuous the previous answer, Theorem 4 and 5 provide a *statistical/approximation analysis* for our learning objective assuming there is no optimization error (as it is usually done in statistical ML).
>
> The theorems tell us that our empirical learning objective is valid from the statistical perspective (error could be bounded by the well-celebrated Rademacher complexity) and the approximation perspective (neural networks can achieve arbitrary small error). Obtaining more precise bounds requires the detailed analysis of weak $C$-transform's statistical properties. This requires significant work for establishing particular theorems, assumptions, definitions. We leave this for the future work.
>
>
> **(4) In addition, from my knowledge of the literature, it does not appear to be common to use neural networks for parameterization of the potentials (correct me if I am wrong)**
>
>
> Please note that we consider the **continuous** OT barycenter problem (see Section 2.1) but  **not discrete**. Here the dual variable is a (continuous) function, not just a finite-length vector. Hence, using the neural-network parameterization for dual potentials is common [1,2]. It not only allows to do the out of sample estimation (which is tricky to do in the discrete case) but also improves the scalability of solvers.
>
>
> **(5) How much computational overhead this will incur on the computation, especially if
>  K is large?**
>
> In accordance with the proposed algorithm on page 6 (see Section 4.2), one can see that the time of the computation will grow linearly. At the same time, we note that some parts of our algorithm could be easily run in parallel. In particular, the MCMC sampling procedures for each $k \in \{1, 2, \dots, K\}$ and computing the $k$-th components of our optimized loss $\widehat{L}_k$ are completely independent. This may significantly improve the computation rates.

---

> > ### Author Response · Authors · 2023-11-21
> > **Answer to Reviewer 16og (2/2)**
> >
> > **(6) The mention of energy-based models (EBMs) does not seem to be very relevant to me either. Ultimately an MCMC procedure such as ULA is used to generate samples in order to approximate an integral. EBMs are related but not that relevant.**
> >
> >
> > At first glance, it might seem that MCMC is the only intersection point of EBMs and our approach. However, upon closer examination of key concepts in EBMs and our proposed solver, much deeper interconnections become apparent, as outlined below.
> >
> > The main concepts of Energy-Based Models (EBMs) are described in accordance with [3, 4] as:
> >
> > (a) **Task:** learning the Energy function (the logarithm of unnormalized density).
> >
> > (b) **Training:** minimization of KL divergence which involves running an MCMC procedure.
> >
> > (c) **Inference:** running MCMC procedure.
> >
> > The corresponding description of our proposed barycenter solver are as follows:
> >
> > (a) **Task:** learning of the dual potentials $f_{k}$ (which together with costs $c_k$ form the logarithm of unnormalized densitites of conditional OT plans).
> >
> > (b) **Training:** minimization of the sum of KL divergences (our Theorem 2) which involves running an MCMC procedure.
> >
> > (c) **Inference:** running an MCMC procedure.
> >
> > Thus, one can seen that our barycenter solver has clear similarities with EBMs training. In fact, it can be viewed as a modification of standard EBM training for solving the barycenter task.
> >
> >
> >
> > **(7) Also, the authors appear to have used ULA with stochastic/batch gradients, which is better known as stochastic gradient Langevin dynamics (SGLD).**
> >
> >
> > We believe there might be a misunderstanding since SGLD is not relevant to the scope of our paper. To see this, let us compare the concepts of SGLD vs. ULA in our solver. In accordance with the initial paper of SGLD [5], the target task and the cause of stochasticity are described as:
> >
> >
> > (a) **Target:** get samples $\theta$ from posterior parameter distribution $p(\theta|X)\propto p(\theta)\prod_{n=1}^{N}p(x_n|\theta)$;
> >
> > (b) **Cause of stochasticity:** batch estimation of gradient $\nabla_{\theta}\log p(\theta|x)\approx \nabla_{\theta}\log p(\theta)+\frac{N}{n}\sum_{i=1}^{n}\nabla_{\theta}\log p(x_{t_{i}}|\theta)$, i.e., using batches $\{x_{t_{1}},\dots,x_{t_{n}}\}$ from the dataset to **perform approximate Langevin steps**.
> >
> > The corresponding elements of our ULA procedure in our solver are as follows:
> >
> > (a) **Target:** get samples $y$ from the $k$-th conditional plan $\pi^{f_{k}}(y|x_{k})$;
> >
> > (b) **Cause of stochasticity:** none. We know $\nabla_{y} \log \pi^{f_{k}}(y|x_{k})$ analytically (Section 4.2): it equals $\nabla_{y}\frac{f_k(y)-c_{k}(x_k,y)}{\epsilon}$, i.e., we perform **exact Langevin steps** by using this gradient;
> >
> > Thus, one can see that ULA procedure in our paper makes sampling from conditional OT plan whose analytically-known gradient is used without batch-estimation (absence of stochasticity) unlike SGLD. But already samples from our ULA procedure is used for Monte-Carlo estimation of the integral (Eq. 11).
> >
> > **(8) Typo: "swap $\text{min}$ and $\text{sup}$ in (17)"**
> >
> > We would like to kindly address a potential misunderstanding. Formula (17) is obtained by substituting dual formulation of EOT (3) in the objective (8). Specifically, we get:
> > $$(8) = \inf_{\mathbb{Q} \in \mathcal{P}(Y)} \sum_{k = 1}^{K} \lambda_k EOT_{c_{k, \epsilon}}(\mathbb{P}_k, \mathbb{Q}) = $$
> >
> > $$ = \inf_{\mathbb{Q} \in \mathcal{P}(Y)} \sum_{k = 1}^{K} \lambda_k \sup_{f_k \in \mathcal{C}(Y)} [\int_{\mathcal{X}} f_{k}^{C}(x) d P_k(x) + \int_{\mathcal{Y}} f_k(y) d \mathbb{Q}(y)  ]=(17)$$
> >
> >
> > **Concluding remarks.** Please reply to our post and inform us if the clarifications provided adequately address your concerns regarding our work. We are more than willing to discuss any remaining points during the discussion phase. If the responses offered meet your satisfaction, we kindly request you to consider raising your score.
> >
> > **Additional references.**
> >
> > [1] Alexander Korotin, Vahe Egizarian, Lingxiao Li and Evgeny Burnaev. Wasserstein Iterative Networks for Barycenter Estimation. In NeurIPS, 2022.
> >
> > [2] Jiaojiao Fan, Amirhossein Taghvaei, Yongxin Chen.Scalable Computations of Wasserstein Barycenter via Input Convex Neural Networks. In ICML, 2021.
> >
> > [3] Yann LeCun, Sumit Chopra, Raia Hadsell, M Ranzato, and Fujie Huang. A tutorial on energy-based
> > learning. Predicting structured data, 1(0), 2006.
> >
> > [4] Yang Song, Diederik P. Kingma. How to Train Your Energy-Based Models. arXiv:2101.03288
> >
> >
> > [5] Max Welling, Yee Whye Teh. Bayesian Learning via Stochastic Gradient Langevin Dynamics. In ICML, 2011.

---

> > > ### Comment · Reviewer_16og · 2023-11-23
> > > **Response**
> > >
> > > Thanks the authors for your answers to my questions and clarified some of my misunderstanding. I think some of the answers like (2) and (3) indicate some of the concerns I have remain valid. But the authors have made comprehensive explanation for their work. I hope the authors would address my concerns about the theoretical contribution of this work. I have raised my score.

---

### Author Response · Authors · 2023-11-21
**General response**

Dear reviewers, thank you for taking the time to review our paper! We are delighted that you positively highlight the novelty of our algorithm (Reviewer 7U49, BAkA), our theoretical insights (Reviewer 7U49) and empirical evidence supporting our theoretical findings (Reviewer 16og, 97mW). Please find the answer to your share question below.

**(1) Comparison with ground-truth entropic barycenters. (Reviewers BAkA, 97mW)**

We note there exist many ways to incorporate the entropic regularization for barycenters [2, Table 1]; they yield non-coinciding barycenter problem statements with different solutions. For some of them, the ground-truth solutions are known for specific cases such as the quadratic cost barycenters in the Gaussian case. Unfortunately, to our knowledge, the Gaussian ground-truth solution for **our** problem statement is not yet known. To address the reviewers' comments, we added a **new Appendix C.4**, where

- We detailed the above mentioned aspects of barycenter problem and cited the suggested paper (BAkA) [1].

- We *quantitatively evaluate* our solver (with small $\epsilon\approx 0$) in recovering the ground-truth *unregularized* ($\epsilon=0$) OT barycenter of Gaussians which can be computed. Our method performs  sometimes even beats the recent baseline solver (which is designed specifically for $\epsilon=0$) considered therein.

**Revised paper.** We have uploaded the paper revision where we fixed typos (pointed by the Reviewers 16og, BAkA), cited the suggested papers (Reviewer BAkA) and added the new Appendix C.4 which is discussed above (Reviewers BAkA, 97mW). **All the edits are highlighted with the blue color.** Please take a look.

**References.**

[1] Khang Le, Dung Le, Huy Nguyen, Dat Do, Tung Pham, Nhat Ho. Entropic Gromov-Wasserstein between Gaussian Distributions. In ICML, 2022.

[2] Chizat, L. (2023). Doubly Regularized Entropic Wasserstein Barycenters. arXiv preprint arXiv:2303.11844.

---

### Meta-Review · Area_Chair_UAip · 2024-01-02

**Metareview:**

This paper proposes a new family of solvers to approximate the Wasserstein barycenter of K measures. These solvers output an unnormalized energy based model (EBM) that one can sample from (using MCMC, e.g. ULA) to obtain points from the barycentric measure. The solver builds upon the dual formulation of the weak OT barycenter problem, which is parameterized by K dual potential functions. The loss to train these dual functions exploits a congruence relationship between the K potentials of a barycenter problem, in addition to a typical dual integral term; combined this results in (11). The loss requires, through the (weak OT) C-transform of EOT potentials, integrating over samples of K EBMs (here the energy is a NN + a cost), which simplifies, as presented in (Mokrov et al 23 Eq. 14) to integrating log-partition functions. The gradient of (11) only involves an expected (samples first produced using ULA, involving derivative of NN functions + costs) gradient of these NN functions using the log-derivative trick. The authors provide experiments in two 2D problems (synthetic + MNIST with 2 digits) as well as a variant of CelebA data.

The paper received four reviews, three of them were of good quality, and pointed out a few issues with the paper (scores of 5,6,6, some of them updated up during the rebuttal to reach these values).

In addition to these 3 reviews, a shorter review liked the paper directly (8) but with little substance. Unfortunately, the latter reviewer (*7U49*) declined twice to substantiate their opinion during the rebuttal phase. Despite me clearly stating I had to discard their review due to their lack of detail, the reviewer confirmed they were ok with this. As a result, I had to dig a bit deeper into the paper, to provide here a semi-review.

Reviewers *BAkA*, *97mW*, *16og* all worry to a varying degree about the incrementality of this work over (Mokrov et al 23), and the feasibility of the approach in more challenging setups since it relies on ULA as an inner routine. A reviewer suggested to extend the approach to unbalanced problems; while this is an interesting direction, that point was not considered to reach a decision.

The issues highlighted by reviewers remained to varying degrees after and during the rebuttal discussion, which I think is reflected in the borderline scores. The paper sits on the fence, and although I think the paper could be an interesting addition, I will side with the reviewers for mostly the following reasons:
- many parts of the paper are pulled (sometimes verbatim) from Mokrov et al, 23. While the paper is pleasant, the reader does feel that the contribution is a specialized reformulation of the EBM framework from Mokrov to the more specific problem of barycenters.
- the experimental validation, while interesting, is mostly limited to CelebA faces, and a preprocessed version of it from [Korotin et al 22a].
- The authors insist on multiple occasions that their methods reaches more generality because it can handle general costs, and therefore contributes to the literature in a different area than previous methods. In experiments, however, all costs are reformulations of the L2^2 cost (pending a symmetric or asymmetric mapping) or use a pre-trained StyleGAN generator to introduce an asymmetric cost (barycenter in latents + data space). It's therefore not clear to me this method would work as a generic/reliable OT barycenter framework for higher dimensional data if it weren't for the stylegan restriction / trick.

Overall the paper has many merits (pushing the EBM narrative in OT is certainly one of them) but is too close to very recent work to meet the bar for a truly novel methodological innovation at ICLR.

Minor comments:
- It would make sense, given the links with weak OT, to add a reference to "A novel notion of barycenter for probability distributions based on optimal weak mass transport" (Cazelles Tobar Fontbona, Neurips'21)
- twister map $u$ is not very clearly defined in R^2 in S5.1 (nor in C.1). Defining it directly on x cartesian coordinates might make it more clear, as well as "Consider *the* map".
- "The particular values of number of steps L and step size η are reported in the details of the experiments, see Appendix C": these numbers vary quite a lot, some discussion on how they were obtained could be useful (e.g. robustness to these choices).
- The authors mention frequently that their method works for general costs (e.g. bottom of p.8) but a discussion on the need for differentiable costs (due to reliance on ULA) could be useful.
- In the gradient step of Algorithm 1, "Perform a gradient step over θ by using ∂L", it would have been more clear to explicitly mention "stop_gradient" steps, as in (Mokrov et al 23) Alg. 1.

**Justification For Why Not Higher Score:**

The paper feels like a direct extension of Mokrov et al' 23, without solving its shortcomings, but mostly highlighting a new application. The experiments are only mildly convincing to justify an accept in that case.

I really believe the 8 should be discarded, leaving this at 5.66 average score.

**Justification For Why Not Lower Score:**

NA

---

### Decision · Program_Chairs · 2024-01-16

Reject